# Rotary Position Encodings for Graphs

**Isaac Reid** [* 1 2]   **Arijit Sehanobish** [* 3]   **Cederik Höfs** [* 1]   **Bruno Mlodozeniec** [1 4]   **Leonhard Vulpius** [1]
**Federico Barbero** [5 2]   **Adrian Weller** [1 6]   **Krzysztof Choromanski** [2]   **Richard E. Turner** [1 6]   **Petar Veličković** [2 1]

## Abstract

We study the extent to which rotary position encodings (RoPE), a recent transformer position encoding algorithm broadly adopted in large language models (LLMs) and vision transformers (ViTs), can be applied to graph-structured data. We find that rotating tokens depending on the spectrum of the graph Laplacian efficiently injects structural information into the attention mechanism, boosting performance in synthetic and real-world graph learning tasks. This approach, coined *Wave-Induced Rotary Encodings* (WIRE), enjoys intriguing theoretical properties: it recovers regular RoPE on grids, and depends asymptotically on the graph effective resistance. Unlike bias-based relative position encodings, WIRE is compatible with linear attention.

## 1. Introduction

Position encodings incorporate information about the respective locations of tokens into the transformer attention mechanism (Vaswani et al., 2017). This is important because the meaning of a sequence of words or image patches depends upon how they are ordered. Likewise, the meaning of a graph depends upon how its constituent nodes are connected. Position encodings capture these spatial and topological relationships, enabling the network to learn expressive functions that generalise well to unseen data.

**APEs and RPEs**. Early transformers relied on *absolute* position encodings (APEs), which add or concatenate fixed or learned embeddings to each token (Kiyono et al., 2021; Liu et al., 2020). Whilst simple, these generally perform worse than *relative* position encodings (RPEs), which instead modulate attention logits for each query-key pair by

---

[*]Core contributors. [1]University of Cambridge [2]Google DeepMind [3]Independent Researcher [4]Max Planck Institute for Intelligent Systems [5]University of Oxford [6]Alan Turing Institute.

*Proceedings of the 43rd International Conference on Machine Learning*, Seoul, South Korea, PMLR 306, 2026. Copyright 2026 by the author(s).

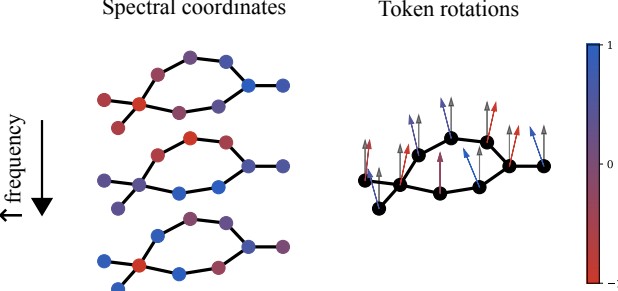

Spectral coordinates    Token rotations

*Figure 1.* **WIRE schematic.** Rotary position encodings (RoPE) (Su et al., 2024) can be applied to graphs using graph spectra. The first few eigenvectors of the Laplacian vary slowly across $\mathcal{G}$; higher frequencies oscillate sharply between adjacent nodes. The eigenvectors are projected down to obtain rotation angles for every query and key. WIRE enjoys desirable theoretical properties and is compatible with linear attention (Choromanski et al., 2021; Katharopoulos et al., 2020).

a bias, taking $\boldsymbol{q}_i^\top \boldsymbol{k}_j \to \boldsymbol{q}_i^\top \boldsymbol{k}_j + b_{ij}$ (Shaw et al., 2018). The bias $b_{ij}$ depends on the tokens' respective positions, e.g. sequence separation in text or shortest path distance between graph nodes. Recent years have witnessed RPEs in turn be superseded by *rotary* position encodings (RoPE) (Su et al., 2024). RoPE decomposes tokens into 2-dimensional blocks and rotates them by position-dependent angles. RoPE's strong empirical performance and modest computational footprint have fuelled its growing popularity in LLMs and ViTs (Dubey et al., 2024; Gemma Team, 2024; Heo et al., 2024). Moreover, it enjoys the convenient property that (as with APEs) it directly modifies tokens, rather than the logits of query-key pairs. This makes RoPE compatible with linear attention and KV-caching, improving scalability.

**Position encodings for graphs**. Without a canonical 'coordinate system', position encodings for *graphs* -– sets of nodes connected by edges -– are more complicated. There are many possible notions of 'position' and 'distance' in a graph. One choice is to use the spectrum of the graph Laplacian to build APEs (Dwivedi and Bresson, 2020; Kreuzer et al., 2021). In the special case of grid graphs, this closely resembles the sinusoidal APEs applied to text and images. Whilst straightforward to implement, such methods do not in general encode invariance properties. Alternatively, one can compute some structural property like the shortest path distance or effective resistance for each pair of graph nodes,

and use these quantities as RPE biases (Ying et al., 2021; Zhang et al., 2023). This approach is more effective than APEs, but requires instantiation of the $N \times N$ attention matrix. It is incompatible with linear attention.

**Core contributions**. Motivated by the shortcomings of APEs and RPEs, and the strong success of RoPE in LLMs and ViTs, this paper asks the following question.

*To what extent can rotary position encodings be applied to graph-structured data?*

We answer via the following core contributions. (1) We introduce **WIRE** (*Wave-Induced Rotary Encodings*), a RoPE-style position encoding for graphs. See Figure 1. (2) We show that WIRE is more general than RoPE, and that it can stochastically downweight attention scores based on graph effective resistance. (3) We demonstrate that WIRE improves transformer performance in synthetic graph tasks, experiments with point clouds, and graph benchmarks, including with linear attention.

## 2. Preliminaries

Consider an undirected graph $\mathcal{G}(\mathcal{N}, \mathcal{E})$, where $\mathcal{N} := \{v_1, ..., v_N\}$ is a set of $N$ nodes and $\mathcal{E}$ is a set of edges. $(v_i, v_j) \in \mathcal{E}$ if and only if there exists an edge between $v_i$ and $v_j$ in $\mathcal{G}$. The number of nodes $N$ is equal to the number of transformer tokens. Let $\{x_i\}_{i=1}^N \subset \mathbb{R}^d$ denote this set of $d$-dimensional tokens. $d$ is assumed to be even.

**Attention**. The $i$th query, key and value vectors are given by $q_i = \mathbf{W}_q x_i$, $k_i = \mathbf{W}_k x_i$ and $v_i = \mathbf{W}_v x_i$ respectively, with $\mathbf{W}_q, \mathbf{W}_k, \mathbf{W}_v \in \mathbb{R}^{d \times d}$ learned projection matrices. For simplicity of notation we assume the single-head setting, with the understanding that all arguments are trivially generalised to multi-head attention. The *attention mechanism*, one of the fundamental computational units of the transformer, is written:

$$x_i \rightarrow \frac{\sum_j \text{sim}(q_i, k_j) v_j}{\sum_{j'} \text{sim}(q_i, k_{j'})}. \quad (1)$$

Here, $\text{sim}(\cdot, \cdot) : \mathbb{R}^d \times \mathbb{R}^d \rightarrow \mathbb{R}$ is a 'similarity' function that assigns a score to each query-key pair. Standard softmax attention uses $\text{sim}(q_i, k_j) = \exp(q_i^\top k_j)$, whereas linear attention takes $\text{sim}(q_i, k_j) = q_i^\top k_j$ (Katharopoulos et al., 2020). The former generally works better, but the latter enables one to write a low-rank decomposition of the attention matrix, unlocking $\mathcal{O}(N)$ scaling. Concretely, with a slight abuse of notation, with linear attention one can take $x_i \rightarrow q_i^\top \left( \sum_j k_j v_j \right) / q_i^\top \left( \sum_{j'} k_{j'} \right)$. The commutativity of matrix-matrix multiplication obviates instantiating the attention matrix $\left[ \text{sim}(q_i, k_j) \right]_{i,j=1}^N \in \mathbb{R}^{N \times N}$ in memory.

In the same spirit, one can define (random) feature maps $\varphi(\cdot) : \mathbb{R}^d \rightarrow \mathbb{R}^m$ and take $\text{sim}(q_i, k_j) = \varphi(q_i)^\top \varphi(k_j)$, again unlocking $\mathcal{O}(N)$ scaling (Choromanski et al., 2021). Common choices for $\varphi(\cdot)$ include ReLU activations and random Laplace features (Yang et al., 2014).

**Rotary position encodings**. Suppose that each token is equipped with a $m$-dimensional coordinate $r_i \in \mathbb{R}^m$, with $m = 1$ for sequences, $m = 2$ for images and $m = 3$ for videos and point clouds. Given a (projected) token $z_i \in \{q_i, k_i\}$, RoPE takes $z_i \rightarrow \text{RoPE}(r_i)z_i$, where:

$$\text{RoPE}(r_i)z_i := \bigoplus_{n=1}^{d/2} \rho(\theta_n)[z_i]_{2n-2:2n-1},$$
$$\rho(\theta) := \begin{pmatrix} \cos(\theta) & -\sin(\theta) \\ \sin(\theta) & \cos(\theta) \end{pmatrix}, \quad \theta_n := \omega_n^\top r_i. \quad (2)$$

Here, $\bigoplus$ denotes the direct product, so each $2 \times 2$ matrix $\rho(\theta_n)$ rotates a 2-element section of the query or key. Meanwhile, $\{\omega_n\}_{n=1}^{d/2} \subset \mathbb{R}^m$ are learnable or fixed frequencies.[1] Using the basic properties of 2D rotations, it is straightforward to see that

$$\text{RoPE}(r_i)^\top \text{RoPE}(r_j) = \text{RoPE}(r_j - r_i), \quad (3)$$

whereupon the joint transformation of queries and keys takes $q_i^\top k_j \rightarrow q_i^\top \text{RoPE}(r_j - r_i)k_j$. Clearly, RoPE is translationally invariant,[2] an inductive bias that helps it generalise to new sequence lengths and makes it effective in 3D robotics applications (Schenck et al., 2025).

**Transformers for graphs**. Whilst Graph Neural Networks (GNNs) have traditionally performed best for graph-structured data, recent years have witnessed growing interest in transformers (Müller et al., 2024; Veličković et al., 2018; Ying et al., 2021). A key algorithmic challenge is to design effective position encodings that capture important structural information about $\mathcal{G}$. To this end, researchers often consider graph spectra (Chung, 1997).

**Graph spectra**. Let us denote the graph *adjacency matrix* by $\mathbf{A} := \left[ \mathbb{I}\left( (v_i, v_j) \in \mathcal{E} \right) \right]_{i,j=1}^N \in \{0, 1\}^{N \times N}$, whose $(i, j)$ entry is equal to 1 if the corresponding edge is present in the graph and 0 otherwise. Let $\mathbf{D} := \text{diag}\left( \sum_j \mathbf{A}_{ij} \right)$ denote the diagonal degree matrix. The *graph Laplacian* is given by $\mathbf{L} := \mathbf{D} - \mathbf{A} \in \mathbb{R}^{N \times N}$. Since it is symmetric, we can write

---

[1] For legibility, we generally suppress the dependence of $\text{RoPE}(r_i)$ on $\{\omega_n\}_{n=1}^{d/2}$, leaving it implicit.

[2] Given this property, some researchers taxonomise RoPE as a type of relative position encoding (RPE). However, we prefer to distinguish it as a separate class of PE, since PEs based on other high-dimensional rotations in $\text{SO}(d)$ are not necessarily translationally invariant (Schenck et al., 2025).

$$\mathbf{L} = \mathbf{U}\mathbf{\Lambda}\mathbf{U}^{\top}, \quad \mathbf{\Lambda} = \mathrm{diag}(\lambda_0, ..., \lambda_{N-1}), \quad (4)$$

with $\lambda_0 \leq \lambda_1 \leq ... \leq \lambda_{N-1}$. Here, the eigenvectors matrix $\mathbf{U} := [\boldsymbol{u}_0, \boldsymbol{u}_1, ..., \boldsymbol{u}_{N-1}]^{\top}$ is orthonormal, with each $\boldsymbol{u}_i \in \mathbb{R}^N$ oscillating across the graph at frequency $\lambda_i$. The spectrum of $\mathbf{L}$ (or $\mathbf{D}^{-1/2}\mathbf{L}\mathbf{D}^{-1/2}$) captures the structure of $\mathcal{G}$. $\mathbf{U}$ and $\mathbf{\Lambda}$ are often used to construct graph transformer APEs. Here, we will use them within RoPE.

**Remainder of the manuscript**. In Section 3 we introduce *Wave-Induced Rotary Encodings* (WIRE), generalising RoPE to graphs. We show that WIRE enjoys a host of attractive theoretical properties. In Section 4, we demonstrate that WIRE performs competitively in learning tasks with a strong structural component.

# 3. WIRE: Wave-Induced Rotary Encodings

How can one apply RoPE to graphs, which lack a canonical coordinate system? Specialising to some particular set of graph-based 'positional features' $\{\boldsymbol{r}_i\}_{i=1}^{N} \in \mathbb{R}^m$ to feed into Eq. (2) implicitly sets an invariance property via Eq. (3), which we want to instill a sensible inductive bias.

Given the efficacy of Laplacian eigenvectors $\{\boldsymbol{u}_i\}_{i=1}^{N}$ as APEs, it is tempting to also try using them as inputs to RoPE. We find this simple approach to work surprisingly well. We formalise this as *Wave-Induced Rotary Encodings* (WIRE) below. Much of the rest of the paper is dedicated to demonstrating and understanding the effectiveness of WIRE in graph learning tasks.

---

**Alg. 1. Wave-Induced Rotary Encodings (WIRE).**

1. Compute the lowest $m \leq N$ eigenvectors and eigenvalues $\{\boldsymbol{u}_k, \lambda_k\}_{k=0}^{m-1}$ of the graph Laplacian $\mathbf{L}$, either exactly or with approximate iterative methods.

2. Define *spectral features* for each graph node, e.g. $\boldsymbol{r}_i = [\boldsymbol{u}_k[i]]_{k=0}^{m-1} \in \mathbb{R}^m$ or similar.

3. Apply rotary position encodings using these spectral features, taking $\boldsymbol{z}_i \rightarrow \mathrm{RoPE}(\boldsymbol{r}_i)\boldsymbol{z}_i$ for queries and keys $\boldsymbol{z}_i \in \{\boldsymbol{q}_i, \boldsymbol{k}_i\}$.

---

We use the word 'wave' to reflect that $\{\boldsymbol{u}_i\}$ capture increasing frequencies of oscillation across $\mathcal{G}$. For example, $\boldsymbol{u}_0$ is constant across a connected graph, whereas $\boldsymbol{u}_1$ (the Fiedler vector) varies slowly and can be used to compute graph partitions into two components. Truncating at $m$ frequencies captures the $m$ slowest oscillations across the graph.

## 3.1. Straightforward observations about WIRE

We begin with the following remarks about Alg. 1, which are simple but important.

**Expressivity of WIRE**. WIRE can distinguish graphs identical under the 1-dimensional Weisfeiler-Lehman graph isomorphism heuristic (with colours replaced by node features), because their adjacency matrices and hence node spectral coordinates differ. In this sense, transformers equipped with WIRE are *more expressive than standard GNNs*, which notoriously fail this test (Morris et al., 2019; Xu et al., 2019). Of course, this simple property is shared by all but the simplest of geometric machine learning models – consider e.g. *higher-order graph neural networks* (Morris et al., 2019), *provably powerful graph neural networks* (Maron et al., 2019) or *Graphormer* (Ying et al., 2021). It is included here as a sanity check.

**Number of parameters**. The only learnable parameters in WIRE are the frequencies $(\boldsymbol{\omega}_i)_{i=1}^{d/2} \subset \mathbb{R}^m$, i.e. $dm/2$ parameters per transformer layer. Typically $m \ll d$, so this is very small compared to the rest of the network. For additional savings, one can share WIRE weights between layers or heads, or even follow conventional RoPE by freezing frequencies in an exponential decay pattern (Su et al., 2024).

**WIRE and GNNs**. In practice, for many graph-based tasks a combination of global attention and message passing layers gives the best performance, rather than a pure transformer (Rampášek et al., 2022; Shirzad et al., 2023). Naturally, WIRE is compatible with such hybrid models; one simply incorporates it wherever attention is used.

**Generalising WIRE**. Lastly, we remark that, whilst Alg. 1 is presented for the special case of *spectral* graph features, one can in principle feed any reasonable node representation into RoPE – e.g. RWPE (Dwivedi et al., 2021) or WavePE (Khang Ngo et al., 2023). We prefer the spectral formulation because we will later see that it admits interesting theoretical analysis, recovering regular RoPE on grid graphs (Theorem 2) and exhibiting asymptotic dependence on graph effective resistance (Theorem 3). We discuss other variants, which may trade these theoretical properties for greater computational efficiency, in Section 4.3 and Appendix A.7.

Next, we discuss more involved theoretical properties.

## 3.2. Node ordering permutation equivariance and recovering RoPE on grids

The following is true.

**Lemma 1:** *The WIRE transformation is equivariant under permutation of the ordering of the nodes of the graph, up to ambiguity of sign flips and rotations in degenerate subspaces.*

*Proof.* The Laplacian is permutation equivariant: reordering the node indices via a permutation matrix $\mathbf{P}$ permutes

the rows and columns of $\mathbf{L}$ in kind, taking $\mathbf{L} \to \mathbf{PLP}^\top$. The eigenvectors and eigenvalues of this *permuted* Laplacian are trivially $\{\mathbf{P}\boldsymbol{u}_k, \lambda_k\}_{k=0}^{N-1}$ – making them respectively equivariant and invariant. The only ambiguity in the eigenvectors comes from the following sources: (1) if $\boldsymbol{u}_i$ is an eigenvector of $\mathbf{L}$, then $-\boldsymbol{u}_i$ is also an eigenvector with the same eigenvalue; (2) if an eigenvalue has multiplicity greater than 1 (common in symmetric graphs like grids and complete graphs), any linear combination of the corresponding eigenvectors is also an eigenvector. That is, eigenvectors are only uniquely defined up to sign flips and rotations in degenerate subspaces. ∎

This basis ambiguity issue is well-studied in geometric deep learning. It is not specific to WIRE. To remedy it, practitioners often apply extra transformations to the spectral features to ensure that they are invariant under sign flips and subspace rotations, or *gauge invariant*. For instance, one might use maximal axis projection (Ma et al., 2023), sign flipping heuristics, or SignNet (Lim et al., 2023). This is trivially incorporated into WIRE, but we find it makes little difference in practice. An alternative strategy, also used in GraphGPS (Rampášek et al., 2022), is to train the transformer on enough data that it 'learns' these ambiguities, mapping inputs with sign-flipped or subspace-rotated eigenvectors to (approximately) the same output. The ambiguities are treated like data augmentation. In our experiments (see Section 4.1 and Section 4.3 later), we find this approach to be sufficient. Moreover, using exact eigenvectors rather than e.g. pushing them through an MLP is often important for theoretical results – for instance, for obtaining asymptotic dependence on effective resistance (Theorem 3). We will also later see that, despite basis ambiguities, the *limiting* transformation of randomised WIRE is actually exactly gauge invariant. This may help explain our method's robustness.

**Recovering RoPE on grids.** Since RoPE works very well with text and image data (Heo et al., 2024; Su et al., 2024), it is desirable for WIRE to recover similar behaviour on the special case of grid graphs. Consider the following.

**Theorem 2:** *RoPE is a special case of WIRE, occurring when one considers a grid graph $\mathcal{G}$ with specific learnable frequencies $\{\boldsymbol{\omega}_n\}_{n=1}^{\frac{d}{2}}$.*

*Proof.* First consider a 1D grid (formally denoted as the path $P_N$), with adjacency matrix $\mathbf{A}_{ij} = \delta_{i,j+1} + \delta_{i,j-1}$. For this specific graph, the second (first nontrivial) eigenvector of $\mathbf{L}$ is given by $\boldsymbol{u}_1 = \left[-\cos\left(\frac{1}{N}\left(i+\frac{1}{2}\right)\pi\right)\right]_{i=0}^{N-1}$. This changes monotonically between $-\cos\left(\frac{\pi}{2N}\right)$ at $i = 0$ and $\cos\left(\frac{\pi}{2N}\right)$ at $i = N - 1$. This sequence of coordinates, increasing as one progresses along $P_N$, is completely analogous to the token position coordinates $[0, 1, ..., N - 1]$. They only differ by

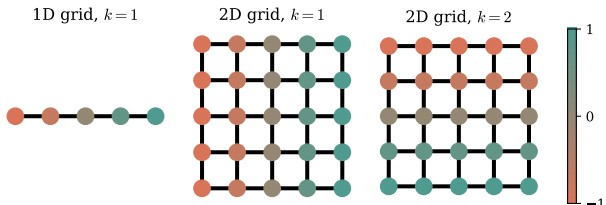

*Figure 2.* **RoPE $\subset$ WIRE.** The leading elements of the Laplacian eigenvectors of grid graphs (formally, Cartesian products of paths $P_N$) change monotonically in each direction. If we apply WIRE using just these coordinates, we recover regular RoPE as used in LLMs and ViTs. In this sense, RoPE is a special case of WIRE.

rescaling by $\frac{\pi}{N}$, offsetting by a constant, and restricting to the range $(-1, 1)$ by pushing through a cosine transformation. Taking $\boldsymbol{\omega}_i = [0, \omega_i, 0, 0, ..., 0]$, we isolate the contribution from this first nontrivial spectral coordinate and recover regular RoPE used in LLMs, up to these simple bijective coordinate system transformations. See Figure 2 left.

Next, consider a 2 dimensional grid graph of size $N_x \times N_y$. This can be expressed as the Cartesian product $P_{N_x} \Box P_{N_y}$, so the spectrum factorises. Completely analogously to the 1D case, the second and third eigenvectors are $\boldsymbol{u}_1[i] = \left[-\cos\left(\frac{1}{N_x}\left(i_x + \frac{1}{2}\right)\pi\right)\right]_{i=0}^{N-1}$ and $\boldsymbol{u}_2[i] = \left[-\cos\left(\frac{1}{N_y}\left(i_y + \frac{1}{2}\right)\pi\right)\right]_{i=0}^{N-1}$, with $i_y = \left\lfloor\frac{i}{N_x}\right\rfloor$ and $i_x = i - N_x i_y$. The order of $\boldsymbol{u}_1$ and $\boldsymbol{u}_2$ will depend on whether $N_x$ or $N_y$ is greater, but this detail is not important.[3] Taking $\boldsymbol{\omega}_i = [0, \omega_x, \omega_y, 0, ...]$, we now recover regular RoPE for ViTs. This is equivalent to applying 1D RoPE for each axis independently. See Figure 2 centre and right.

These arguments generalise to higher-dimensional grids (e.g. 3D for video), where one considers products of a progressively greater number of path graphs. ∎

Theorem 2 demonstrates that regular RoPE and WIRE on (products of) path graphs are closely related, which helps build confidence that WIRE will be effective in graph learning tasks. It provides another helpful sanity check. It is interesting to interrogate the minor differences between WIRE on path graphs and regular RoPE.

1. First, as noted above, the spectral coordinates are always normalised to the range $(-1, 1)$, rather than taking values $0, ..., N - 1$. This type of coordinate renormali-

---

[3]Likewise, if e.g. $N_x \gg N_y$, $\boldsymbol{u_2}$ as written may no longer correspond specifically to the *second* smallest nonzero eigenvalue; it may occur later on in the ordering. This technical detail obfuscates our presentation and is not of substantive importance. The point is that oscillations purely in the $y$ direction correspond to one of the modes of $\mathcal{G}$ and can be isolated with suitable $\boldsymbol{\omega}$.

sation is actually a popular trick in LLMs to improve generalisation with respect to sequence length (Chen et al., 2023; Li et al., 2024). It is intriguing that WIRE incorporates this regularisation automatically. We posit that it might improve generalisation to different graph sizes.

2. Second, since the eigenvectors of $P_N$ are only unique up to a sign, one could equally flip the direction of all the spectral coordinates. This is not a property exhibited by RoPE when used in LLMs – here, there is a clear sense of directionality. Parity invariance follows from the fact that we consider undirected $\mathcal{G}$, so it is to be expected.

Lemma 1 and Theorem 2 discussed the simplest theoretical properties of WIRE – equivariance under node ordering permutation, and recovering RoPE on grids. Next, we explore WIRE's more involved invariance guarantees, proving asymptotic dependence on effective resistance.

### 3.3. WIRE depends on effective resistance

Recall that the commutativity and orthogonality of 2D rotations make RoPE translationally invariant: $(\text{RoPE}(\boldsymbol{r}_i)\boldsymbol{q}_i)^\top \text{RoPE}(\boldsymbol{r}_j)\boldsymbol{k}_j = (\text{RoPE}(\boldsymbol{r}_i + \boldsymbol{c})\boldsymbol{q}_i)^\top \text{RoPE}(\boldsymbol{r}_j + \boldsymbol{c})\boldsymbol{k}_j \ \forall \ \boldsymbol{c} \in \mathbb{R}^m$. To rephrase, the composite transformation $\text{RoPE}(\boldsymbol{r}_j - \boldsymbol{r}_i)$ applied to a query-key pair (implicitly in the case of linear attention) only depends upon the tokens' *separation* $\boldsymbol{r}_j - \boldsymbol{r}_i$, rather than their absolute positions. This property is found to be important in 3D robotics applications (Schenck et al., 2025). It also helps sequence length generalisation in LLMs (Peng et al., 2023; Su et al., 2024).

WIRE automatically inherits the property described above. However, the interpretation of translational invariance in *spectral* space is less clear. Invariance under shortest path distance – a popular choice for RPE schemes made e.g. in Graphormer (Ying et al., 2021) – might be more intuitive.

A closely-related alternative to shortest path distance is the *effective resistance* (Ellens et al., 2011; Velingker et al., 2023; Zhang et al., 2023), defined by

$$R(i,j) := \mathbf{L}^\dagger_{ii} + \mathbf{L}^\dagger_{jj} - 2\mathbf{L}^\dagger_{ij} \tag{5}$$

for nodes $(i,j) \in \mathcal{N}^2$. Here $\mathbf{L}^\dagger$ is the Laplacian pseudoinverse, which removes any diverging component in the regular inverse in the zero eigenvalue direction. It is straightforward to confirm that $R(i,j)$ is a metric on $\mathcal{N}^2$. It is also known that effective resistance provides a lower bound for shortest path distance, with equality achieved for trees (Spielman, 2010).

Interestingly, when one applies randomised WIRE with spectral features, one implicitly modulates query-key logits

depending upon the respective nodes' effective resistance. This is formalised with the following result.

**Theorem 3:** *Consider a connected graph with spectral features* $\boldsymbol{r}_i = \left[\boldsymbol{u}_k[i]/\sqrt{\lambda_k}\right]_{k=1}^{N-1} \in \mathbb{R}^{N-1}$. *Suppose that we randomly sample the WIRE frequencies* $\boldsymbol{\omega}_i \sim \mathcal{N}(0, \omega\mathbf{I}_{N-1})$, *with* $i = 1, ..., \frac{d}{2}$ *and* $\omega \in \mathbb{R}$. *Given a query-key pair* $(\boldsymbol{q}_i, \boldsymbol{k}_j) \in \mathbb{R}^d \times \mathbb{R}^d$, *we have that*

$$\mathbb{E}\left[(\text{RoPE}(\boldsymbol{r}_i)\boldsymbol{q}_i)^\top \text{RoPE}(\boldsymbol{r}_j)\boldsymbol{k}_j\right] = \\ \boldsymbol{q}_i^\top \boldsymbol{k}_j (1 - \omega^2 R(i,j)/2) + \mathcal{O}(\omega^4), \tag{6}$$

*where* $R(i,j)$ *is the effective resistance between nodes* $i, j \in \mathcal{N}$. *That is, in expectation, the leading contribution of WIRE is to downweight query-key logits by a factor proportional to the effective resistance.*

*Proof.* For connected graphs, $\mathbf{L}^\dagger = \sum_{k=1}^{N-1} \frac{1}{\lambda_k} \boldsymbol{u}_k \boldsymbol{u}_k^\top$ since $\lambda_0 = 0$ but $\lambda_k \neq 0$ for $k \geq 1$. It is straightforward to see that $R(i,j) = \sum_{k=1}^{N-1} \frac{1}{\lambda_k}(\boldsymbol{u}_k[i] - \boldsymbol{u}_k[j])^2$. For each node $i \in \mathcal{N}$, we define an $N-1$-dimensional spectral feature $\boldsymbol{r}_i = \left[\boldsymbol{u}_k[i]/\sqrt{\lambda_k}\right]_{k=1}^{N-1} \in \mathbb{R}^{N-1}$, whereupon $R(i,j) = \|\boldsymbol{r}_i - \boldsymbol{r}_j\|_2^2$. Considering random weights[4] $\boldsymbol{\omega}_i \sim \mathcal{N}(0, \omega\mathbf{I}_{N-1})$,

$$\mathbb{E}\left((\boldsymbol{\omega}^\top \boldsymbol{r}_i - \boldsymbol{\omega}^\top \boldsymbol{r}_j)^2\right) = \omega^2 \|\boldsymbol{r}_i - \boldsymbol{r}_j\|_2^2. \tag{7}$$

Given a query-key pair $(\boldsymbol{q}, \boldsymbol{k})$ at positions $(\boldsymbol{r}_i, \boldsymbol{r}_j)$,[5]

$$\boldsymbol{q}^\top \boldsymbol{k} \to \boldsymbol{q}^\top \text{RoPE}(\boldsymbol{r}_i)^\top \text{RoPE}(\boldsymbol{r}_j)\boldsymbol{k} =$$

$$\sum_{k=1}^{\frac{d}{2}} (\boldsymbol{q}_{2k-2}\boldsymbol{k}_{2k-2} + \boldsymbol{q}_{2k-1}\boldsymbol{k}_{2k-1})\cos(\boldsymbol{\omega}_k^T(\boldsymbol{r}_i - \boldsymbol{r}_j)) \tag{8}$$

$$+ (\boldsymbol{q}_{2k-1}\boldsymbol{k}_{2k-2} - \boldsymbol{q}_{2k-2}\boldsymbol{k}_{2k-1})\sin(\boldsymbol{\omega}_k^T(\boldsymbol{r}_i - \boldsymbol{r}_j)).$$

Taylor expanding in $\omega$ and taking the expectation,

$$\mathbb{E}(\boldsymbol{q}_i^\top \boldsymbol{k}_j) \to \boldsymbol{q}_i^\top \boldsymbol{k}_j \left(1 - \frac{\omega^2}{2}\|\boldsymbol{r}_i - \boldsymbol{r}_j\|_2^2\right) + \mathcal{O}(\omega^4)$$

$$= \boldsymbol{q}_i^\top \boldsymbol{k}_j \left(1 - \frac{\omega^2}{2}R(i,j)\right) + \mathcal{O}(\omega^4) \tag{9}$$

as claimed. Here, we used the fact that sin is an odd function to drop the $\mathcal{O}(\omega^3)$ terms. ∎

Theorem 3 shows that, under certain assumptions, the limiting WIRE transformation is related to effective resistance.

---

[4]This is nothing other than the celebrated Johnson-Lindenstrauss transformation (Dasgupta and Gupta, 2003), a random projection that preserves vector norms and distances in expectation.

[5]In Eq. (8), we drop the $i$ and $j$ suffixes on the queries and keys for compactness, freeing it up to represent the coordinate $k \in \{0, ..., d-1\}$.

However, we stress that WIRE is not *exactly* invariant under effective resistance for a particular set of frequencies $(\boldsymbol{\omega}_i)_{i=1}^{d/2}$ due to (1) the $\mathcal{O}(\omega^4)$ correction terms and (2) the requirement of the expectation $\mathbb{E}(\cdot)$. In practice, one does not sample and average over an ensemble of random WIRE transformations, but instead takes one learnable instantiation. Nonetheless, this result helps build intuition for how WIRE modulates the attention between pairs of nodes: the further apart they are, the more attention tends to be downweighted.

**Why is this result interesting?** The remarkable feature of Theorem 3 is that the limiting invariance guarantee holds *without needing to instantiate the attention matrix in memory*. To wit, one can (approximately) modulate the attention matrix entry as

$$\boldsymbol{q}_i^\top \boldsymbol{k}_j \rightsquigarrow \boldsymbol{q}_i^\top \boldsymbol{k}_j (1 - \omega^2 R(i,j)/2), \qquad (10)$$

but without explicitly computing all $N^2$ scores $\{\boldsymbol{q}_i^\top \boldsymbol{k}_j\}_{ij=1}^N$ or resistances $\{R(i,j)\}_{ij=1}^N$. Previously, such invariance properties have only been possible via expensive *relative* position encodings (RPEs), which instantiate and soft-mask the whole attention matrix. Unlike RPEs, WIRE – which encodes invariances by *separately* rotating queries and keys – is compatible with linear attention. This type of principled 'linear attention topological masking' has long been a goal of the efficient transformer research community (Chen et al., 2023; Choromanski et al., 2022; Reid et al., 2024).

**Theorem 3 is gauge invariant**. Another interesting observation is that Theorem 3 *still holds* under random sign flips and basis transformations of the eigenvectors. The leading term in Eq. (9) depends upon $\|\boldsymbol{r}_i - \boldsymbol{r}_j\|_2^2$, which is unmodified when these modifications are applied to $\{\boldsymbol{r}_i\}_{i=1}^N$. To rephrase, whilst Lemma 1 caveats WIRE's permutation equivariance under node reordering, the fundamental asymptotic behaviour of (randomised) WIRE does **not** depend upon these ambiguities in basis and sign. The limiting WIRE transformation is actually intrinsically gauge invariant. This may help explain why WIRE still works well in our experiments (see Section 4.1 and Section 4.3), without instilling exact gauge invariance via e.g. SignNet (Lim et al., 2023). Absolute position encodings do not in general enjoy this property.

The various theoretical contributions discussed so far may be succinctly summarised as follows.

> **Takeaways from WIRE theory**. When considering the special case of grid graphs (formally, Cartesian products of path graphs $P_N$), WIRE recovers regular RoPE as used in LLMs and ViTs. If we instantiate WIRE with random weights, then the expected limiting transforma-

> tion can downweight query-key logits depending upon their effective resistance – a lower bound to shortest path distance. Remarkably, WIRE exhibits this behaviour *without* needing to explicitly instantiate the attention matrix in memory, so it is compatible with linear attention.

We finish this section by discussing WIRE's time complexity, and techniques to speed it up.

### 3.4. Efficient instantiations of WIRE

The RoPE transformation in Eq. (2) is *extremely* efficient to compute. This is because the full RoPE matrix is blockwise $2 \times 2$ and thus very sparse. Explicitly, in view of Eq. (2), one can simply take:

$$\begin{aligned}
\boldsymbol{z}_i \rightarrow &\left[\cos(\theta_1), \cos(\theta_1), ..., \cos\left(\theta_{\frac{d}{2}}\right), \cos\left(\theta_{\frac{d}{2}}\right)\right] \odot \boldsymbol{z}_i \\
&+ \left[-\sin(\theta_1), \sin(\theta_1), ..., -\sin\left(\theta_{\frac{d}{2}}\right), \sin\left(\theta_{\frac{d}{2}}\right)\right] \odot \mathbf{P}\boldsymbol{z}_i.
\end{aligned} \qquad (11)$$

Here, $\odot$ denotes the Hadamard (element-wise) product. $\mathbf{P} := \left[\delta_{\lfloor i/2 \rfloor, \lfloor j/2 \rfloor} - \delta_{i,j}\right]_{i,j=0}^{d-1} \in \{0,1\}^{d \times d}$ is the permutation that takes $\mathbf{P}\boldsymbol{x} = [\boldsymbol{x}_1, \boldsymbol{x}_0, \boldsymbol{x}_3, \boldsymbol{x}_2, ..., \boldsymbol{x}_{d-1}, \boldsymbol{x}_{d-2}]$, swapping alternate vector entries. Eq. (11) only needs $\mathcal{O}(d)$ operations.

On the other hand, one may be more concerned by the $\mathcal{O}(N^3)$ time complexity to exactly diagonalise $\mathbf{L}$. We do not think this is a problem in practice for the following reasons.

1. There exists a rich literature on efficient approximate diagonalisation techniques for big graphs – e.g. coarsening (Loukas and Vandergheynst, 2018) and the Lanczos method (Lanczos, 1950). The latter is especially relevant since for WIRE one often only uses the lowest few eigenvectors, not the whole spectrum. See Halko et al. (2011) for a helpful overview. Many such methods are $\mathcal{O}(N)$ in the size of the original graph.

2. One can also depart from graph spectra entirely, applying WIRE using features which are cheaper to compute like FastRP (Chen et al., 2019) and *random walk position encodings* (Dwivedi et al., 2021). See Appendix A.7 for a demonstration. This loses our theoretical guarantees, but can perform well in practice.

3. Practitioners nearly always *already* compute some structural node feature to be used as an APE. This amortises the precomputation cost; the same features can also be applied via WIRE, at negligible extra overhead.

## 4. Experiments

Here, we test WIRE on a range of graph-based tasks, training $> \mathbf{200}$ **transformer models** in total. WIRE provides

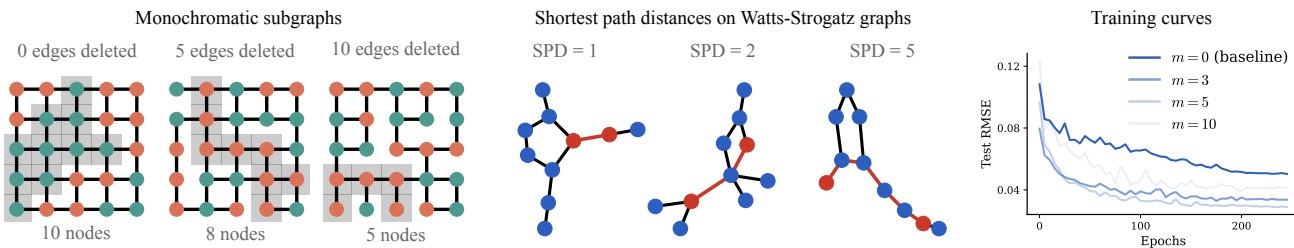

*Figure 3.* **Synthetic experiments**. *Left*: Predict the number of nodes in the largest connected monochromatic subgraph(s) (shaded). Varying numbers of edges are removed, shown above. *Center*: Random graphs labelled with shortest path distances between target and source nodes (red). *Right*: Corresponding training curves with $m \in \{0, 3, 5, 10\}$ spectral features.

a strong topological inductive bias, which often boosts performance.

### 4.1. Synthetic tasks: monochromatic subgraphs and shortest paths

**Synthetic task 1: Monochromatic subgraphs.** We begin with a synthetic task, chosen to strongly depend upon the structural properties of $\mathcal{G}$. We generate $10,000$ train graphs and $1,000$ test graphs with $N = 25$ nodes, beginning with a $5 \times 5$ grid and then deleting a randomly selected subset of edges. Every node is assigned a colour. We train a transformer to predict the size, i.e. number of nodes, of the largest monochromatic connected subgraph(s). See Figure 3 left.

The motivation for constructing graphs as described above is that changing the number of deleted edges allows us to interpolate between 2D grid graphs and more complicated topologies. For grids WIRE can recover RoPE (Theorem 2), which is already known to perform well. The setup is similar to a ViT. On the other hand, as we delete more edges the topology becomes more complicated, testing how WIRE fares with trickier $\mathcal{G}$.

For the model inputs, we use the Laplacian eigenvectors, concatenated with node colour labels. This means that all our models include APEs by default. For WIRE, we use spectral features using variable $m \in \{0, 3, 5, 10\}$. Clearly, $m = 0$ corresponds to *not* using WIRE. Growing $m$ incorporates progressively higher-frequency structural information into the rotations. Full architecture and training details are in Appendix A.1.

Normalised test RMSEs are shown in Table 1, with standard errors in parentheses. WIRE provides gains over the baseline model ($m = 0$). When $\mathcal{G}$ is close to a grid, low-dimensional spectral features are sufficient. In contrast, as we delete more edges and $\mathcal{G}$ becomes more complicated, higher frequencies become helpful.

**Synthetic task 2: Predicting shortest path distances.** Next, we generate random Watts-Strogatz graphs with $N = 10$ nodes and $k = 2$ neighbours, with rewiring probability

*Table 1.* **Monochromatic subgraph task**. Normalised test RMSEs for computing the largest monochromatic connected subgraph. $m$ is the spectral coordinate dimensionality; WIRE is used wherever $m > 0$. WIRE substantially improves regression performance.

| | Test RMSE (↓) | | | |
|---|---|---|---|---|
| | Num. deleted edges | | | |
| $m$ | 0 | 5 | 10 | 15 |
| 0 (no WIRE) | 0.060(1) | 0.087(1) | 0.081(1) | 0.068(2) |
| 3 | **0.053(2)** | 0.075(2) | 0.072(3) | 0.064(3) |
| 5 | 0.057(2) | 0.075(1) | 0.070(2) | **0.056(4)** |
| 10 | 0.055(2) | **0.068(5)** | **0.063(2)** | 0.058(2) |

*Table 2.* **Shortest path distance task**. WIRE provides strong improvements to transformers trained to predict shortest path distances on random Watts-Strogatz graphs.

| | Num. spectral coords, $m$ | | | |
|---|---|---|---|---|
| | 0 (no WIRE) | 3 | 5 | 10 |
| **Test RMSE (↓)** | 0.065(5) | 0.048(6) | **0.038(6)** | 0.045(4) |

$p = 0.6$. Again, we take $10,000$ training examples and $1,000$ test examples. We train transformer models, in the same way as before, to predict the shortest path distance (SPD) between two randomly selected nodes. Figure 3 gives three examples, with the target and source nodes indicated in red and the corresponding SPD labelled above.

Given WIRE's dependence on resistive distance (Theorem 3) – a lower bound to SPD – we expect it to provide a strong inductive bias. Table 2 confirms that this is indeed the case; WIRE nearly halves the test RMSE compared to the APE-only baseline ($m = 0$). Figure 3 *(right)* shows sample training curves. Appendix A.1 gives further experimental details.

**Number of parameters**. In the experiments above, the WIRE parameters constitute a tiny fraction of the entire model: **less than 1%** when $m = 3$. It is remarkable that they nonetheless lead to a strong performance boost. This spectral information is *already* being fed into the model as inputs. WIRE simply converts this into an additional strong

structural inductive bias, applied throughout the network at every layer and every attention head.

### 4.2. Point cloud transformers

Next, we consider point cloud data (Guo et al., 2021). To implement WIRE, we construct a sparse $k$-nearest neighbours graph. The input features are $(x, y, z)$ for each point. For economy of space, details and results are in Appendix A.3 (Table 4). WIRE provides consistent gains over the regular PCT baseline, for both softmax and linear attention.

### 4.3. WIRE Performers on GNN benchmark tasks

Finally, we evaluate WIRE on established graph benchmarking tasks. To showcase its compatibility with linear attention, we focus on $\mathcal{O}(N)$ Performer models. Here, $\mathcal{O}(N^2)$ bias-based RPE methods *cannot* be applied because we do not instantiate the full attention matrix. See Table 3.

**WIRE is a drop-in addition to existing models**. For a clean, competitive implementation, we incorporate WIRE into GraphGPS architectures known to perform well on benchmarking tasks (Rampášek et al., 2022). These are idiosyncratic; the best combination of message passing, attention and MLPs depends upon the task at hand. We use ReLU linear attention. Full details are in Appendix A.4.

Remarkably, across the board, adding WIRE – a lightweight, extra structural inductive bias – can improve performance by multiple points. In the context of these already heavily-optimised tasks, the gains are substantial. Whilst the linear variant still often performs worse than its expensive full-rank counterpart (the price of greater efficiency), we observe that WIRE is frequently able to *substantially close this gap*. For instance, on MalNet-Tiny, WIRE Performers are just as effective as transformers, but unlike the latter we can train on a single T4 12GB GPU.

**WIRE beyond Performers**. WIRE can be used within *any* model applying attention on $\mathcal{G}$. For example, WIRE often also provides gains when used with $\mathcal{O}(N^2)$ softmax attention, as noted in Section 4.1 and Section 4.2. We give further examples for a subset of the GNN benchmark datasets (smaller instances, where poor scalability is not prohibitive) in Table 8 of Appendix A.4. Equally, WIRE can be used within other efficient transformers like SGFormer (Wu et al., 2023) and BigBird (Zaheer et al., 2020) (the latter combined with GNNs within GPS), again improving test accuracy. See Table 9 in Appendix A.4. These short demonstrations provide further evidence of WIRE's broad utility. We defer exploration with other variants – such as Exphormers (Shirzad et al., 2023), which use virtual global nodes and expander graphs (Deac et al., 2022; Wilson et al., 2025), and Graph Attention Networks (Veličković et al., 2018), which use local attention – as important future work.

*Table 3.* **Graph benchmarks**. Performer test metrics with/without WIRE, on graph benchmarks. (↑)/(↓) indicates whether higher or lower scores are better. For comparison, less efficient $\mathcal{O}(N^2)$ baselines from Rampášek et al. (2022) are also shown in gray. Instances where WIRE manages to close the Performer-softmax gap (to within experimental noise) are shown in red.

| Dataset | Performer $\mathcal{O}(N)$ | | Transformer $\mathcal{O}(N^2)$ |
|---|---|---|---|
| | Baseline | WIRE | Baseline |
| MNIST (↑) | 97.56(2) | **98.10(1)** | 98.05(4) |
| CIFAR10 (↑) | 70.61(4) | **71.15(3)** | 72.3(1) |
| PATTERN (↑) | 85.71(3) | **86.63(6)** | 86.69(2) |
| CLUSTER (↑) | 76.90(3) | **77.53(3)** | 78.02(6) |
| ogbg-molhiv (↑) | 0.776(2) | **0.785(2)** | 0.788(1) |
| ogbg-molpcba (↑) | 0.238(3) | **0.264(1)** | 0.291(3) |
| ogbg-ppa (↑) | 0.8009(8) | **0.804(2)** | 0.802(3) |
| ogbg-code2 (↑) | 0.1731(9) | **0.1733(9)** | 0.189(2) |
| Peptides-func (↑) | 64.4(1) | **64.9(1)** | 65.4(4) |
| Peptides-struct (↓) | 0.2616(4) | **0.2566(4)** | 0.2500(5) |
| PascalVOC-SP (↑) | 0.367(1) | **0.376(1)** | 0.37(1) |
| MalNet-Tiny (↑) | 92.81(5) | **93.46(2)** | 93.36(6) |

**WIRE with random walk position encodings**. Lastly, we again stress that, whilst WIRE with *spectral* coordinates enjoys interesting theoretical properties, it can in principle be applied with any efficient graph node representation. We give examples using *random walk position encodings* (RWPEs) (Dwivedi et al., 2021) in Appendix A.7, again finding accuracy gains. This could unlock further speedups compared to spectral features – for sparse graphs, one can compute RWPEs in $\mathcal{O}(N)$ time. It also suggests robustness to the choice of RoPE feature. See Table 11 and Table 12 for full details. We remark that RWPE WIRE provides an interesting inductive bias: pairs of nodes that have similar 'local environments', but are not necessarily close together in $\mathcal{G}$, experience less attention modulation.

> **Takeaways from WIRE experiments**. Just as RoPE improves the performance of LLMs and ViTs, WIRE provides a structural inductive bias that boosts the accuracy of transformers on graph-structured data. This includes in synthetic settings, as well as GNN benchmarks. In some cases, WIRE appears to reduce or even fully close the gap between linear and softmax attention.

## 5. Conclusion

We studied the extent to which rotary position encodings (RoPE) – a popular algorithm previously limited to LLMs and ViTs – can be applied to graphs. Using graph spectra to

rotate tokens, we introduced *Wave-Induced Rotary Encodings* (WIRE). Unlike many graph position encodings – e.g. in Graphormer (Ying et al., 2021) – WIRE is compatible with linear attention since one does not need to instantiate the attention matrix. In experiments, we find WIRE to be effective in tasks where a strong structural inductive bias is important. We hope this work will prompt further research generalising RoPE to new data modalities like graphs.

## Impact Statement

This paper presents work whose goal is to advance the field of machine learning. There are many potential societal consequences of our work, none of which we feel must be specifically highlighted here. We have made every effort to ensure the work's reproducibility. The core algorithm is presented clearly in Alg. 1. Theoretical results are proved with accompanying assumptions in the main body. Code is available here: https://github.com/cederikhoefs/Graph-RoPE. It builds upon existing public repositories. The datasets in Section 4.3 are standard and freely available online. Exhaustive experimental details about the training and architectures are reported in Appendix A.

## Acknowledgements and Contributions

IR gratefully acknowledges funding from Trinity College, Cambridge and from Google. CH thanks Studienstiftung des deutschen Volkes and German Academic Exchange Service for their financial support. AW acknowledges support from EPSRC via a Turing AI Fellowship under grant EP/V025279/1, and from the Leverhulme Trust via CFI. RET is supported by the EPSRC Probabilistic AI Hub (EP/Y028783/1).

IR and FB independently suggested generalising rotary position encodings to graphs. Together with PV, they jointly supervised a mini-project by LV and CH. CH continued to work on this during a master's project, jointly supervised by IR and PV. For his thesis, CH ran initial tests of the method on real and synthetic tasks. IR wrote the text, proved all theoretical results, and ran the experiments in Section 4.1. AS ran the GNN experiments in Section 4.3, with contributions from CH and BM. AS also ran the various extra ablations in the Appendices. KC, RET, AW and PV provided helpful high-level guidance throughout, including feedback on the manuscript.

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

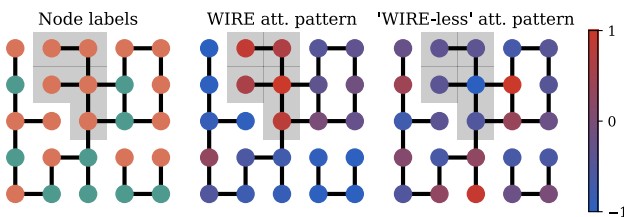

*Figure 4.* **Example attention patterns with WIRE.** Random choice of model input *(left)*, and example attention patterns for a trained model with *(centre)* and without *(right)* WIRE. WIRE helps nodes attend to other nearby nodes with the same label.

# A. Extra experimental details

In this appendix, we provide extra experimental details to supplement Section 4.

### A.1 Synthetic experiments: monochromatic subgraphs and shortest paths

**Models and training**. For both tasks, we use a standard 4 layer transformer with model and MLP dimensionality 32. For simplicity, the attention is single-head. We train for 250 epochs with batch size 16, with a learning rate of $2 \times 10^{-4}$ obeying a cosine decay schedule ($\alpha = 0.01$). We train using the Adam optimiser with weight decay $1 \times 10^{-4}$. Dropout is applied at a rate of 0.2 to attention and the MLP outputs. Graph embeddings are obtained by mean pooling over node embeddings, and a dense layer projects the result to a scalar prediction for (1) the size of the largest monochromatic connected subgraph and (2) the shortest path distance between a target and source node (identified at the model inputs). Both datasets have 10, 000 training examples and 1, 000 test examples. We report the lowest test root mean squared error obtained during training, normalised by graph size. Standard errors are computed over 4 runs per setting.

**Ablation: WIRE attention patterns**. To better understand WIRE, we can also examine the activations of a trained model. For instance, Figure 4 shows rescaled attention scores at the final layer of the network. We take identical optimised weights, with WIRE either switched on as during training *(centre)* or off *(right)*. With WIRE, we see that nodes attend within the biggest monochromatic subgraph. The pattern disappears when WIRE is removed. This suggests that the network does indeed learn to use query-key rotations to carry structural information about $\mathcal{G}$.

### A.2 WIRE and Performers

Recall that, for $\mathcal{O}(N)$ Performer attention, we take:

$$x_i \to \frac{\sum_j \varphi(q_i)^\top \varphi(k_j) v_j}{\sum_{j'} \varphi(q_i)^\top \varphi(k_{j'})}, \tag{12}$$

where $q_i = \mathbf{W}_q x_i, k_i = \mathbf{W}_k x_i$ and $v_i = \mathbf{W}_v x_i$ respectively, with $\mathbf{W}_q, \mathbf{W}_k, \mathbf{W}_v \in \mathbb{R}^{d \times d}$ learned projection matrices. $\varphi(\cdot) : \mathbb{R}^d \to \mathbb{R}^m$ is a (random) feature map, common choices for which include ReLU activations and random Laplace features (Yang et al., 2014).

There are two obvious manners in which one could incorporate WIRE:

1. *Directly modulating the queries and keys.* $z_i \to \text{RoPE}(r_i) z_i$ for $z_i \in \{q_i, k_i\}$.
2. *Modulating the features.* $\varphi(z_i) \to \text{RoPE}(r_i) \varphi(z_i)$ for $z_i \in \{q_i, k_i\}$.

The benefit of (1) is that, for suitable choices of maps $\varphi(\cdot)$ like ReLU, we have that

$$\varphi(\text{RoPE}(r_i) q_i)^\top \varphi(\text{RoPE}(r_j) k_j) \geq 0. \tag{13}$$

The attention scores all remain positive, which avoids instabilities caused by the denominator changing sign. Conversely, the advantage of (2) is that

$$(\text{RoPE}(r_i) \varphi(q_i))^\top (\text{RoPE}(r_j) \varphi(k_j)) = \varphi(q_i)^\top \text{RoPE}(r_j - r_i) \varphi(k_j), \tag{14}$$

*Table 4.* **PCT results**. Test accuracy with different position encodings for classification and segmentation tasks, including both regular and efficient (Performer) attention. WIRE is consistently best (**boldface**) or second best (underlined), achieving greater accuracy than the regular PCT baseline (NoPE). It performs similarly to Cartesian RoPE, using $(x, y, z)$.

| | Test accuracy ($\uparrow$) | | | |
| --- | --- | --- | --- | --- |
| | Classification (ModelNet40) | | Segmentation (ShapeNet) | |
| **PE** | Transformer | Performer | Transformer | Performer |
| NoPE | 91.8 | 90.1 | 93.1 | 92.8 |
| Cart. RoPE | 91.8 | **90.8** | **93.2** | **93.2** |
| WIRE | **93.4** | **90.8** | **93.2** | 93.0 |

which gives desirable invariance properties. But now modulated attention scores *can* be negative which can in general cause instabilities – something that Su et al. (2024) sidestep by only applying RoPE to the numerator (see Eq. 19 of their paper).

In Performer experiments, we find (1) to work well in practice, so tend to adopt this approach.

### A.3 Point cloud transformers

To begin, note the following motivation for using WIRE with PCTs: point cloud WIRE is invariant under SE(3) transformations. Trivially, the nearest neighbours graph $\mathcal{G}$ is invariant under joint translation and rotation of the point cloud data – namely, SE(3) transformations. The same follows for its spectrum, and thus the WIRE transformation we apply to queries and keys. Conversely, this property does *not* hold for RoPE transformations with 3D Cartesian coordinates, where rotation and translation will in general modify the position encoding.

**Classification and segmentation**. We train transformer models for classification and semantic segmentation, on the ModelNet40 (Sun et al., 2022) and ShapeNet (Chang et al., 2015) datasets respectively. Each example has 2048 points. We test (1) regular softmax attention, and (2) ReLU linear attention (a 'Performer') (Choromanski et al., 2021). For WIRE, we use spectral features of dimensionality $m = 10$. The nearest neighbours graphs are constructed taking $k = 20$, which gives connected, sparse $\mathcal{G}$. As baselines, we include regular transformer and Performers without any additional position encoding ('NoPE'), as well as regular RoPE using Cartesian coordinates ('Cart. RoPE').

**Results**. The classification test metric is the precision of the object-level predictions (top one correctly classified). For semantic segmentation, it is the accuracy of the point-level predictions, weighted by the number of each type of point. Table 4 gives the results. Runs are expensive, so following standard practice we report a single seed (Guo et al., 2021; Qi et al., 2017). WIRE outperforms the regular PCT (NoPE) baseline for both transformers and Performers, and often matches or surpasses Cartesian RoPE.

For classification, we consider the ModelNet40 dataset (Sun et al., 2022). Each includes 2048 points and belongs to one of 40 object classes, including 'airplane', 'chair' and 'sofa'. The goal is to predict these labels. Meanwhile, for semantic segmentation we consider ShapeNet (Chang et al., 2015). Each point has an associated 'part label', breaking the object up into between 2 and 6 smaller semantically-meaningful sections – e.g. the legs or seat of a chair. The goal is to predict the class labels of each point.

**Models and training**. Building on the Scenic codebase (Dehghani et al., 2022),[6] we use a 4-layer transformer with hidden and MLP dimensions 128 and 512 respectively, trained for $10,000$ epochs with batch size 1024. We experiment with incorporating WIRE into only a subset of layers, anticipating that early layers that capture geometric information will benefit more from improved position encodings than the later semantic layers. This hyperparameter is optimised by a sweep. As baselines, we include regular transformer and Performers without any additional position encoding (NoPE), as well as regular RoPE using Cartesian coordinates (c.f. spectral). We train with the Adam optimiser, with weight decay 0.01. The learning rate schedule is compound (constant, cosine decay and linear warmup) with $10,000$ warmup steps and a base rate of $5 \times 10^{-6}$.

### A.4 GNN benchmark hyperparameters

---

[6]See especially https://github.com/google-research/scenic/tree/main/scenic/projects/pointcloud.

*Table 5.* **Graph benchmark datasets**. Statistics of the datasets considered in Section 4.3.

| Dataset | # Graphs | Avg. nodes | Avg. edges | Dir. | Level / Task | Metric |
|---|---|---|---|---|---|---|
| MNIST | 70,000 | 70.6 | 564.5 | Yes | Graph, 10-class cls. | Accuracy |
| CIFAR10 | 60,000 | 117.6 | 941.1 | Yes | Graph, 10-class cls. | Accuracy |
| PATTERN | 14,000 | 118.9 | 3,039.3 | No | Inductive node, binary cls. | Accuracy |
| CLUSTER | 12,000 | 117.2 | 2,150.9 | No | Inductive node, 6-class cls. | Accuracy |
| ogbg-molhiv | 41,127 | 25.5 | 27.5 | No | Graph, binary cls. | AUROC |
| ogbg-molpcba | 437,929 | 26.0 | 28.1 | No | Graph, 128-task cls. | Avg. Precision |
| MalNet-Tiny | 5,000 | 1,410.3 | 2,859.9 | Yes | Graph, 5-class cls. | Accuracy |
| PascalVOC-SP | 11,355 | 479.4 | 2,710.5 | No | Inductive node, 21-class cls. | F1 score |
| Peptides-func | 15,535 | 150.9 | 307.3 | No | Graph, 10-task cls. | Avg. Precision |
| Peptides-struct | 15,535 | 150.9 | 307.3 | No | Graph, 11-task regression | MAE |

In this section, we provide training details and hyperparameters for the GNN experiments reported in Section 4.3. We follow the setup of Rampášek et al. (2022). We choose MNIST, CIFAR-10, PATTERN and CLUSTER from 'benchmarking GNNs' (Dwivedi et al., 2020), Peptides-func, Peptides-struct and PascalVOC from the Long Range Graph Benchmark (Dwivedi et al., 2022), and ogbg-molhiv, ogbg-molpcba, ogbg-ppa and ogbg-code2 from the OGB datasets (Hu et al., 2020). We also consider MalNet-Tiny (Freitas et al., 2021). We provide the statistics for each dataset in Table 5.

We follow the standard train/validation/test split in each case. For all datasets in 'benchmarking GNNs' and OGB – namely, MNIST, CIFAR-10, PATTERN, CLUSTER, ogbg-molhiv, ogbg-ppa and ogbg-molpcba – we run 10 seeds. Since MalNet-Tiny runs are expensive, we run 3 seeds. Likewise, the LRGB datasets – Peptides-func, Peptides-struct and PascalVOC-SP – are replicated 4 times. Lastly, all ogbg-code2 runs were repeated with 6 seeds. We use the AdamW optimiser (Loshchilov and Hutter, 2019) for all our experiments.

Our code is based on PyTorch Geometric. All experiments are run on a T4 GPU, with the exception of ogbg-ppa and ogbg-code2. The latter two datasets are much more compute intensive, and were run on an NVIDIA A100 (80GB) GPU. The results for the baseline dense transformer are taken from Rampášek et al. (2022), while the results for all other baselines are obtained from our own runs. The RoPE computation in Equation (13) is implemented using a learnable linear layer, transforming the spectral coordinates to dimensionality $d/2$. We control the scale of its initialisation with an additional hyperparameter.

A.4.1 GraphGPS experiments: extra details

In this subsection, we provide further implementation details for all experiments using GraphGPS .

The ReLU-Performer model is described in Appendix A.2. For all our experiments, we default to the hyperparameters used by Rampášek et al. (2022). It is well established that performance is highly sensitive to the choice of hyperparameters for each dataset. For ogbg-ppa and ogbg-code2, all the hyperparameter settings were identical to (Rampášek et al., 2022, Table A.3), with optional 16 Laplacian positional encoding dimension for the WIRE Performer. We give details in Table 6.

Finally, following standard practice, for datasets like MNIST, PATTERN, MalNet-Tiny and ogbg-molhiv, we use random walks to provide global structural information. We use 16 walks for MalNet-Tiny and MNIST, and 20 walks for PATTERN and ogbg-molhiv. We also experiment with regular softmax and BigBird attention (Zaheer et al., 2020). In these cases, we again use the same hyperparameters. Details are provided below.

A.4.2 SGFormer experimental details

SGFormer is another efficient transformer architecture, based upon a single linear attention layer and a single message passing layer (Wu et al., 2023). In contrast to our other Performer experiments, SGFormer takes the nonlinearity $\varphi(\cdot)$ to be the identity map. For message passing, we use a GCN. As usual, WIRE is injected into the attention mechanism of the transformer. Again, we mostly revert to the GraphGPS hyperparameters, avoiding extensive tuning to ensure our results are robust. Table 7 gives details.

**A.5 Extra results for other attention mechanisms on GNN benchmarks**

*Table 6.* **GraphGPS Experiments with Performer Attention**. Hyperparameters used for our GraphGPS Experiments

| Hyperparameters | MNIST | CIFAR-10 | PATTERN | CLUSTER | Peptides-struct | Peptides-func | Pascal-Voc | MalNet-Tiny | ogbg-molhiv |
|---|---|---|---|---|---|---|---|---|---|
| Hidden Dim | 64 | 64 | 64 | 48 | 96 | 96 | 96 | 64 | 64 |
| Heads | 4 | 4 | 4 | 8 | 4 | 4 | 8 | 4 | 4 |
| Attention Dropout | .5 | .5 | .5 | .5 | .5 | .5 | .5 | .5 | .5 |
| MPNN | GINE | GatedGCN | GINE | GatedGCN | GatedGCN | GatedGCN | GatedGCN | GatedGCN | GINE |
| # Layers | 3 | 3 | 6 | 16 | 4 | 4 | 4 | 6 | 10 |
| GNN Dropout | .1 | 0. | 0. | .1 | .1 | .1 | .1 | 0. | 0. |
| Learning Rate | 0.0001 | .001 | 0.0005 | 0.0005 | .0003 | .0003 | .0005 | .001 | .0001 |
| Weight Decay | 1e-4 | 1e-5 | 1e-5 | 1e-5 | 0. | 1e-5 | 0. | 1e-5 | 1e-4 |
| # Laplacian Eigenvectors | 16 | 8 | 16 | 10 | 10 | 10 | 10 | 16 | 8 |
| # RWSE Features | 8 | - | 16 | - | - | - | - | 8 | 8 |
| Scheduler | ReduceLR | cos decay | cos decay | cos decay | cos decay | cos decay | cos decay | cos decay | ReduceLR |
| Batch Size | 64 | 64 | 32 | 32 | 128 | 128 | 32 | 4 | 32 |
| Laplacian Position Encoding Dim | - | 16 | - | 16 | 16 | 16 | 16 | - | - |
| Epochs | 150 | 150 | 100 | 100 | 200 | 150 | 300 | 150 | 100 |

*Table 7.* **SGFormer Experiments**. Hyperparameters used for the SGFormer Experiments.

| Hyperparameters | MNIST | CIFAR-10 | PATTERN |
|---|---|---|---|
| Hidden Dim | 128 | 256 | 128 |
| Heads | 2 | 1 | 8 |
| Attention Dropout | .5 | .5 | .5 |
| # GNN Layers | 3 | 2 | 3 |
| GNN Dropout | .1 | .1 | .1 |
| Learning Rate | 0.001 | .001 | 0.0005 |
| Weight Decay | 1e-5 | 0 | 1e-5 |
| # WIRE Features | 16 | 8 | 10 |
| Scheduler | ReduceLR | cosine decay | cosine decay |
| Epochs | 150 | 100 | 150 |
| Batch Size | 32 | 64 | 32 |

*Table 8.* **WIRE results on softmax transformers**. Ablation results for WIRE on $\mathcal{O}(N^2)$ regular transformer architectures, on smaller datasets where poor scalability is not a problem. As observed in Section 4.1, our algorithm still improves performance.

| Dataset | Variant | Test metric | |
|---|---|---|---|
| | | Baseline | WIRE |
| MNIST (↑) | Softmax transformer | 98.05(4) | **98.46(3)** |
| CIFAR-10 (↑) | Softmax transformer | 72.3(1) | **73.48(7)** |
| PATTERN (↑) | Softmax transformer | 86.69(2) | **86.75(2)** |
| CLUSTER (↑) | Softmax transformer | 78.02(6) | **78.19(2)** |
| ogbg-molhiv (↑) | Softmax transformer | 0.788(1) | **0.798(2)** |

Here, we report extra WIRE results with different (non-Performer) architectures, referenced in Section 4.3 of the main text. Specifically, we report results with regular softmax attention, SGFormer (Wu et al., 2023), and BigBird (Zaheer et al., 2020).

The SGFormer architecture is described above in Appendix A.4.2. Meanwhile, BigBird (Zaheer et al., 2020) combines local and global attention. It uses a small fixed number of global tokens that attend to all $N$ tokens. Remaining tokens attend to their neighbours. Table 8 and Table 9 shows that WIRE can be easily integrated these attention mechanisms, boosting the respective baselines.

*Table 9.* **WIRE results on extra efficient transformers**. Ablation results for WIRE on different $\mathcal{O}(N)$ transformer architectures: namely, SGFormer (Wu et al., 2023) and BigBird (Zaheer et al., 2020). Once more, WIRE can provide gains.

| | | Test metric | |
|---|---|---|---|
| **Dataset** | **Variant** | Baseline | WIRE |
| MNIST ($\uparrow$) | SGFormer | 96.78(4) | **97.3(1)** |
| CIFAR-10 ($\uparrow$) | SGFormer | 60.43(8) | **61.36(6)** |
| PATTERN ($\uparrow$) | SGFormer | 85.2(1) | **85.9(1)** |
| MNIST ($\uparrow$) | BigBird | 97.20 | **98.04** |
| CIFAR10 ($\uparrow$) | BigBird | 85.04 | **85.86** |

*Table 10.* **GNN benchmarks $m$ ablation**. Intermediate values of $m$ give the best downstream performance, striking a good balance between the amount of structural information in the features and the strength of the inductive bias.

| **Dataset** | $m = 0$ | $m = 4$ | $m = 8$ | $m = 16$ | $m = 32$ |
|---|---|---|---|---|---|
| MNIST ($\uparrow$) | $97.564 \pm 0.018$ | $97.730 \pm 0.081$ | $97.790 \pm 0.074$ | $\mathbf{98.104 \pm 0.014}$ | $97.880 \pm 0.120$ |
| CIFAR-10 ($\uparrow$) | $70.606 \pm 0.039$ | $71.030 \pm 0.102$ | $\mathbf{71.145 \pm 0.026}$ | $70.860 \pm 0.045$ | $70.680 \pm 0.065$ |
| PATTERN ($\uparrow$) | $85.710 \pm 0.033$ | $86.018 \pm 0.142$ | $\mathbf{86.684 \pm 0.088}$ | $86.628 \pm 0.064$ | $85.940 \pm 0.062$ |
| CLUSTER ($\uparrow$) | $76.896 \pm 0.028$ | $77.114 \pm 0.029$ | $\mathbf{77.714 \pm 0.064}$ | $77.241 \pm 0.204$ | $76.848 \pm 0.047$ |

*Table 11.* **Shortest path distance task with RWPEs**. WIRE provides improvements to transformers trained to predict shortest path distances on random Watts-Strogatz graphs, using RWPEs instead of spectral features.

| | Num. RWPE coords, $m$ | | | |
|---|---|---|---|---|
| | 0 (baseline) | 3 | 5 | 10 |
| **Test RMSE ($\downarrow$)** | 0.061(1) | 0.060(1) | 0.059(1) | **0.055(2)** |

### A.6 GNN benchmarks $m$ ablation

Here, we provide companion results to Table 1 and Table 2 for the GNN benchmarks, showing how performance varies with the number of spectral coordinates $m$ used for the rotation features. As observed in synthetic tasks, intermediate values of $m$ provide the best performance, balancing the amount of structural information with the strength of the inductive bias. Table 10 shows the results. Studying how this behaviour relates to other graph properties such as homophily/heterophily may make for interesting future work.

### A.7 RWPE WIRE

In this section, we demonstrate how WIRE (RoPE for graphs) can also be applied using *random walk position encodings*, rather than graph spectra. **Importantly, this scalable variant does *not* require any $\mathcal{O}(N^3)$ matrix diagonalisation operations, and is fully compatible with $\mathcal{O}(N)$ linear attention – even taking into account precomputation cost**.

**RWPEs**. Considering an adjacency matrix $\mathbf{A}$ and a degree matrix $\mathbf{D}$, the random walk transition matrix is $\mathbf{P} := \mathbf{D}^{-1}\mathbf{A}$. The RWPE feature for node $i$ is

$$\mathrm{RWPE}(v_i) := \left[\mathbf{P}_{ii}, \mathbf{P}_{ii}^2, \mathbf{P}_{ii}^3, ..., \mathbf{P}_{ii}^k\right] \in \mathbb{R}^k, \tag{15}$$

computing the probability of a random walk returning to node $v_i$ after $\{1, 2, ..., k\}$ steps. RWPEs are popular for APEs in the literature (Dwivedi et al., 2021; Rampášek et al., 2022). One can use RWPEs as rotational features for RoPE.

**Shortest path prediction**. Table 11 shows corresponding results (analogous to Table 2) for shortest path prediction, training for 100 epochs. WIRE using graph spectra tends to perform better (and is in general more expensive), but we also observe a gain over the no-WIRE baseline using RWPEs. As in the main text, RWPEs are additionally provided as APEs, isolating the gains from RoPE rotations.

**RWPE WIRE on GNN benchmarks**. Meanwhile, Table 12 shows companion results to Table 3, choosing a representative subset of the benchmark datasets for economy of compute. The hyperparameters are the same as previously. We again find

*Table 12.* **GNN benchmarks using RWPE coordinates**. Companion results to Table 3 with RWPE WIRE, a cheaper variant that does not require Laplacian diagonalisation.

| Dataset | Baseline | WIRE |
|---|---|---|
| MNIST (↑) | $97.56 \pm 0.018$ | $\mathbf{98.16 \pm 0.064}$ |
| CIFAR-10 (↑) | $70.61 \pm 0.039$ | $\mathbf{71.09 \pm 0.102}$ |
| PATTERN (↑) | $85.71 \pm 0.033$ | $\mathbf{86.66 \pm 0.024}$ |
| CLUSTER (↑) | $76.90 \pm 0.028$ | $\mathbf{77.01 \pm 0.10}$ |
| MalNet-Tiny (↑) | $92.81 \pm 0.054$ | $\mathbf{93.33 \pm 0.034}$ |

that WIRE – in this case, the efficient RWPE variant that does not require any $\mathcal{O}(N^3)$ diagonalisation operations – can provides gains over the baseline Performer models.

