# OpenReview forum: "Rotary Position Encodings for Graphs"
_ICML.cc/2026/Conference — ICML 2026 spotlight_

### Official Review · Reviewer_DTRF · 2026-03-09

**Soundness:** 3
**Presentation:** 3
**Significance:** 2
**Originality:** 3
**Overall Recommendation:** 4
**Confidence:** 3

**Summary:**

This paper explores extending rotary position encodings (RoPE), widely adopted in large language models (LLMs) and vision transformers (ViTs), to graph-structured data by proposing the Wave-Induced Rotary Encodings (WIRE) method. By rotating the query and key vectors according to the spectrum of the graph Laplacian, WIRE efficiently injects the graph’s topological positional information into the attention mechanism, thereby significantly improving performance on both synthetic tasks and real-world graph learning tasks. Its main contributions are that WIRE uses the m lowest low-frequency eigenvectors of the Laplacian as positional coordinates for each node to drive block-wise rotations of the queries and keys, making it a more general form of RoPE; it fully degenerates to standard RoPE on grid graphs; and, in the random frequency setting, the expected attention scores are quadratically attenuated based on the effective resistance between nodes (naturally suppressing attention to distant nodes) without the need to explicitly compute any distance matrix, which makes WIRE perfectly compatible with linear attention mechanisms and demonstrates elegant theoretical properties.

**Compliance With Llm Reviewing Policy:**

Affirmed.

**Final Justification:**

The authors have addressed most of my concerns, so I will keep my positive score.

**Key Questions For Authors:**

(1)For graph tasks with strong heterogeneity or tasks that heavily rely on high-frequency structural information, relying solely on the lowest-frequency Laplacian eigenvectors to generate rotation angles in WIRE can indeed lead to non-negligible loss of important structural details.Please provide theoretical or experimental evidence to support your conclusion.


(2)The choice of m theoretically dictates the granularity of the captured graph structure. To better evaluate the trade-off between computational efficiency and model performance, could you provide a performance evolution curve as $m$ increases (e.g., m in {4, 8, 16, 32, 64}) on real-world datasets? This analysis is crucial for assessing the practical "cost-benefit" ratio of the proposed method.

(3)The current WIRE framework primarily focuses on simulating distance-decay biases induced by low-frequency signals. When dealing with heterophilic graphs that require capturing localized high-frequency oscillatory patterns, does the rotary mechanism of WIRE still maintain a significant advantage over conventional Absolute Position Encodings (APE)? Furthermore, is the WIRE architecture flexible enough to support the parallel integration of multi-band spectral information (e.g., simultaneously utilizing eigenvectors from both ends of the spectrum)?

(4)Does the RWPE-based version of WIRE still preserve the theoretical properties outlined in Theorem 2 (reduction to grid RoPE) and Theorem 3 (correlation with effective resistance)? If these theoretical derivations no longer hold for RWPE, how should we unifiedly interpret the performance gains across these two fundamentally different types of feature inputs?

**Limitations:**

yes

**Strengths And Weaknesses:**

Strengths：

(1)This paper addresses a key challenge in graph transformers: positional encoding. Positional encodings are a core bottleneck in transformers, particularly for graph data, which lacks a canonical ordering. The work tackles this pain point head-on, effectively resolving the fundamental tension between expressive power and computational efficiency in graph transformers.

(2)This paper originally combines RoPE with Laplacian spectral features through the WIRE method, representing a creative fusion of existing positional encoding approaches (APE and RPE). It effectively eliminates key limitations of prior RPE methods, most notably their incompatibility with linear attention mechanisms.

(3)The Wave-Induced Rotary Encodings (WIRE) method proposed in this paper offers broad applicability and lightweight efficiency. It can be seamlessly integrated into a wide range of graph transformer architectures, delivering meaningful performance gains with only a very small number of additional learnable parameters.

Weaknesses：

(1)The WIRE method proposed in this paper includes idealized assumptions in some of the boundary conditions of its theoretical derivations. Although the paper acknowledges sign flips and rotational ambiguities of the eigenvectors, it primarily relies on data augmentation to address them rather than enforcing rigorous invariance. Under such conditions, the model’s performance on extreme out-of-distribution (OOD) graph data has not been sufficiently discussed or experimentally validated.

(2)Although WIRE achieves linear complexity during both inference and attention computation, the preprocessing stage—computing exact eigenvectors of the graph Laplacian—requires O(N³) time complexity.

(3)Although WIRE achieves linear complexity during both inference and attention computation, it heavily relies on precomputed graph spectral features. If the graph topology changes dynamically (e.g., through node/edge insertions or deletions), the expensive eigendecomposition must be recomputed.

(4) This research is primarily based on symmetric graph Laplacian matrices, making it suitable for undirected graphs. For common directed graphs such as social networks or citation networks, the Laplacian matrix is no longer symmetric and its spectral analysis becomes more complex; thus, it remains unclear whether WIRE can be directly generalized or maintain its advantageous properties.

---

> ### Author Rebuttal · Authors · 2026-03-30
>
> We warmly thank the reviewer for reading the paper, and for their positive feedback. We are happy to address all questions and concerns below.
>
> 1. **Invariance assumptions**. Theorem 3 serves as an *existence proof* — it helps motivate why WIRE works well in practice, but it isn’t directly/exclusively responsible for the empirical gains we see in our experiments. Importantly, WIRE still works well with truncated eigenvectors, learned frequencies, and even RWPE rotations, all of which lie beyond Thm 3’s assumptions. As discussed in the paper, in order to enforce “rigorous invariance”, practitioners could use (1) tricks like SignNet on the Laplacian eigenvectors, or (2) RWPE coordinates within WIRE.
> 2. **OOD experiments**. Could the reviewer kindly clarify what type of OOD experiments they have in mind? Given its nice extrapolation properties (see line 221), WIRE may well be more robust OOD than APEs.
> 3. **Precomputation cost**. We do not believe the eigenvector computation cost to be a problem for the reasons in Sec 3.4. To remind the reader: (1) there exist the cheaper alternatives to exact Laplacian diagonalisation like RWPEs that work well, and (2) some structural node features are nearly always already computed for APEs and these can also be also used for WIRE at low extra overhead.
> 4. **Directed graphs**. Thanks for the interesting question. We agree that WIRE using eigenvectors is best suited to symmetric adjacency matrices which can be readily diagonalised. For directed graphs, other node embeddings like RWPEs may be more suitable. We have added comments; thanks.
> 5. **“For graph tasks with strong heterogeneity or tasks that heavily rely on high-frequency structural information, relying solely on the lowest-frequency Laplacian eigenvectors to generate rotation angles in WIRE can indeed lead to non-negligible loss of important structural details”**. We agree that truncating the eigenvectors loses high-frequency information. This information may instead be incorporated via other PEs and by message passing layers. To be clear, WIRE is *not* intended to replace the other PEs/message passing; rather, it provides a cheap *extra* inductive bias  that boosts performance. For instance, even the first nontrivial (Fiedler) eigenvector provides useful information about which nodes are located nearby; other parts of the model will be needed to capture higher frequency details. Thanks.
> 6. **$m$ ablation**. We agree that it may be interesting to also include the $m$ ablation shown in Tables 1 and 2 for real world examples, rather than just the synthetic tasks. As such, we have now added the **new results shown below**. As with the synthetic tasks, the lowest few frequencies provide most of the gains, which may be intuitive given Thm 2. Thanks for prompting this addition.
> |Dataset|m=0|m=4|m=8|m=16| m=32|
> |:-|:-|:-|:-|:-|:-|
> |CiFAR| 70.606±0.039 |$\underline{71.030}$±0.102 |**71.145**±0.026|70.860±0.045 |70.680± 0.065 |
> |Cluster| 76.896±0.028 |77.114±0.029| **77.714** ± 0.064 | $\underline{77.241}$ ± 0.204 | 76.848 ± 0.047 |
> |MNIST| 97.564±0.018 |97.730±0.081| 97.790 ± 0.074 | **98.104** ± 0.014 | $\underline{97.880}$ ± 0.120 |
> |Pattern| 85.710±0.033 |86.018±0.142| **86.684** ± 0.088 | $\underline{86.628}$ ± 0.064 | 85.940 ± 0.062 |
>
> 7. **“does the rotary mechanism of WIRE still maintain a significant advantage over conventional Absolute Position Encodings (APE)?”** We respectfully clarify that WIRE should not be considered an *alternative* to APEs, but rather as an *addition*. We agree that investigating  the types of graphs/tasks for which WIRE provides the biggest gains (eg heterophilic/homophilic) is an interesting direction for future work. We have added comments on this; thanks!
> 8. **“Furthermore, is the WIRE architecture flexible enough to support the parallel integration of multi-band spectral information (e.g., simultaneously utilizing eigenvectors from both ends of the spectrum)?”** Yes, if included in the features used to compute the rotation angles, WIRE can incorporate information from both ends of the spectrum. We will make this clear in the text.
> 9. **“Does the RWPE-based version of WIRE still preserve the theoretical properties outlined in Theorem 2 (reduction to grid RoPE) and Theorem 3 (correlation with effective resistance)?”** No, the results rely on using spectral features for rotations, so they would not hold with RWPEs. In this case, pairs nodes with more “similar neighbourhoods” -- and hence RWPE features -- would experience less attention modulation, irrespective of their actual separation/effective resistance in the graph. The RWPE version of WIRE thus gives a different inductive bias, which also appears helpful in our preliminary experiments (see Tables 10 and 11). Thanks for prompting this interesting discussion, which we have added to the paper.
>
> Thanks once more. In light of these clarifications and **extra experiments**, we hope the reviewer will consider raising their score.

---

> > ### Author Rebuttal · Reviewer_DTRF · 2026-04-01
> >
> > Thank you for your detailed response, which has addressed most of my initial concerns. Since I originally provided a positive evaluation, and the authors have clarified that WIRE serves as an additional inductive bias rather than a replacement for APEs, I will maintain my positive score.
> >
> > Regarding OOD, my core concern lies in structural distribution shifts. Specifically, when transferring across datasets with greatly different spectral densities or homophily ratios, it remains unclear whether the rotation parameters in WIRE can retain consistent physical meaning and structural interpretation.

---

> > > ### Author Response · Authors · 2026-04-02
> > >
> > > Thanks for the speedy response, and for marking your concerns as 'fully resolved'!
> > >
> > > Thanks also for the clarification re OOD data and structural distribution shifts. Please note the following.
> > > 1. Thm 3 holds *regardless of the spectral density*, and of course regardless of the homophily ratio (of which WIRE is independent). In this sense, the 'physical meaning' of WIRE -- (randomised) attenuation of attention in proportion to resistive distance -- stays the same across datasets.
> > > 2. Also, the fact that eigenvector elements are restricted to the range (-1,1) for *any* graph means that rotation angles are **automatically upscaled/downscaled** to the appropriate range upon graph distributional shift, in close analogy to 'interpolation' in LLMs [1] which was introduced to improve OOD performance (specifically, context window extension).
> > >
> > > For these reasons, we think WIRE will perform well OOD.
> > >
> > > We have made the above clearer in the paper. We agree that including a short empirical demonstration of robustness (e.g. shortest path prediction for Watts-Strogatz graphs with changing rewiring probability $p$) may help build intuition for the reader. This feels ambitious for such a short rebuttal period, and we respectfully note that we already train $>200$ transformer models (including extras requested by the reviewer). However, we think this experiment is a good idea and plan to add it to a (possible) camera ready version. Thanks for the nice suggestion!
> > >
> > > ___
> > > [1] Functional Interpolation for Relative Positions Improves Long Context Transformers, Li et al., ICLR 2024

---

### Official Review · Reviewer_szZp · 2026-03-11

**Soundness:** 3
**Presentation:** 2
**Significance:** 2
**Originality:** 2
**Overall Recommendation:** 3
**Confidence:** 4

**Summary:**

The paper try to use rotary position encoding in the graph domain.

**Compliance With Llm Reviewing Policy:**

Affirmed.

**Key Questions For Authors:**

1. The are many good position encoding in the graph, like spectral enoding etc. This is not strong motivation of why we need the RoPE in the LLM domain to the graph. The Rope can extend easily to long context. Do we have the similar issue in the graph domain?
2. RopE + linear attention is not novel, as it is well studied in LLM domain.
3. The author implement the idea of 200 transformer model, are those all use same position encoding? and the peroformance of the WIRE don't outperfom the original transformer as show in table 3.

**Limitations:**

yes

**Strengths And Weaknesses:**

Strength:
1. The paper try to use the rotary position eoncding to the graph domain, which is interesting. It extends the applicability of a widely used position encoding method to a new domain, enabling transformers to better capture graph topology.
2. The paper rovides a thorough theoretical foundation for WIRE, including its properties like permutation equivariance and dependence on effective resistance.
3. the authors conduct extensive experiments across synthetic tasks, point cloud data, and graph benchmarks, demonstrating the practical effectiveness of WIRE

Weakness:
1. The are many good position encoding in the graph, like spectral enoding etc. This is not strong motivation of why we need the RoPE in the LLM domain to the graph. The Rope can extend easily to long context. Do we have the similar issue in the graph domain?
2. RopE + linear attention is not novel, as it is well studied in LLM domain.
3. The author implement the idea of 200 transformer model, are those all use same position encoding? and the peroformance of the WIRE don't outperfom the original transformer as show in table 3.

---

> ### Author Rebuttal · Authors · 2026-03-30
>
> We thank the reviewer for reading the text. We are pleased that they find it to be interesting, theoretically thorough, and extensive in its experiments. All questions and concerns are addressed below.
>
> 1. **“The are many good position encoding in the graph, like spectral enoding etc. This is not strong motivation of why we need the ROPE in the LLM domain to the graph.”** We agree that position encodings for graphs are important and well-studied. We are interested in RoPE for graphs because — (1) RoPE is very effective in LLMs but has never before been applied to graphs, and (2) unlike many RPEs, RoPE is compatible with linear attention. Property (2) is especially unusual for graphs, where many graph PEs need instantiation of the full attention matrix.  Lastly, we stress that WIRE is not an alternative to existing PEs, but rather an addition. We hope this makes the paper’s motivation clearer.
> 2. **“The Rope can extend easily to long context. Do we have the similar issue in the graph domain?”** Yes, we think that WIRE will extend effectively to “long contexts” — in this setting, meaning bigger graphs — because of the properties of the spectrum of the graph Laplacian. Please see the discussion around line 221. Thanks for the interesting question.
> 3. **“RopE+ linear attention is not novel, as it is well studied in LLM domain.”** The novelty of this paper is applying RoPE *to graphs* for the first time, and theoretically/empirically studying its properties. We agree that RoPE has previously been applied with linear attention for text/images, but we don’t think this has been done with graphs.
> 4. **“The author implement the idea of 200 transformer model, are those all use same position encoding?”** No. We train over 200 transformer models in total across all our experiments, including GNN benchmarks, PCTs and synthetic tasks. Each baseline has a different set of existing PEs depending on the task at hand, and in each case we assess the gains from also adding WIRE on top. For each setup, all the PE details are provided in the main body or appendix.
> 5. **“peroformance of the WIRE don't outperfom the original transformer”**. WIRE + Performer consistently beats *Performer* baseline, to which it should be compared. Interestingly, it also closes the gap with/matches the more expensive *softmax transformer* baseline. Our WIRE + Performer method is not expected to beat pure softmax, since the latter scales quadratically cf linearly with graph size. However, please do note that softmax + WIRE does tend to outperform pure softmax in experiments -- see Table 8. This is a fairer comparison.
>
> We again thank the reviewer. Having made the clarifications above and rectified some points of minor misunderstanding, we respectfully ask that they consider raising their score. We will be very happy to answer any remaining questions or concerns.

---

> > ### Author Rebuttal · Reviewer_szZp · 2026-04-02
> >
> > 1. I still feel that the paper is trying to use the LLM positional encoding to solve the graph problem. The motivation is not so strong.
> > 2. "WIRE + Performer is not expected to beat pure softmax" — but then why should practitioners use it? The practical value proposition is unclear. If the selling point is linear scaling, the paper should demonstrate this on graphs large enough that softmax is infeasible, not on benchmarks where softmax works fine.

---

> > > ### Author Response · Authors · 2026-04-02
> > >
> > > Thank you for the quick response. We are happy to clarify these points.
> > >
> > > 1. **'I still feel that the paper is trying to use the LLM positional encoding to solve the graph problem.'** This paper is about finding ways to generalise RoPE -- a hugely popular position encoding for sequences, and the topic of *many* recent NeurIPS, ICML and ICLR papers [1] [2] [3] [4] [5] -- to graph-structured data. Generalising RoPE to new data modalities involves a lot of empirical and theoretical work. We also note that our algorithm is **strictly more general than RoPE**, reducing to RoPE in the special case of grid graphs (Thm 2). To us, the fact that RoPE has previously been very successful in LLMs makes it *more* interesting in the context of generalising it to new data modalities, rather than less interesting. What do you think?
> > > 2. **'Why should reviewers use WIRE, if WIRE + Performer does not beat softmax?'** Please consider the following.
> > > -  *WIRE improves each base algorithm*. WIRE + Performer beats Performer (Table 3), and WIRE + softmax beats softmax (Table 8). As such, in both cases, WIRE is seen to improve the base algorithm, which is our goal. Across modalities, it is very standard in the literature for cheap $O(N)$ transformers to perform worse than expensive $O(N^2)$ models [6] [7] [8] [9]. Papers typically focus on suggesting *improvements* to $O(N)$ models to help close the gap.
> > > - *WIRE + Performer does actually sometimes beat/match softmax*. Notwithstanding the above, we think it is remarkable that in many cases WIRE + Performer *does* beat/match base softmax -- namely, on MNIST, PATTERN, ogbg-molhiv, ogbg-ppa, PascalVOC, and MalNET-Tiny. These make up ~ half of the datasets considered in Table 3.
> > > - *Efficiency gains and big graphs*. Finally, please note that we *do* actually test Performer WIRE on graphs that are so big that, with our GPUs, base softmax is not feasible. Please see e.g line 416, which reads: 'For instance, on MalNet-Tiny, WIRE Performers are just as effective as transformers, but unlike the latter we can train on a single T4 12GB GPU'. This **directly shows the scalability benefits of our method**, achieving competitive performance on graphs that are (on our hardware) too big for regular softmax. We hope this clarifies our 'value proposition'.
> > >
> > > We trust that this answers any remaining questions about (1) the motivation of the paper and (2) why $O(N)$ models will not *in general* beat $O(N^2)$ models -- though with Performer + WIRE they actually sometimes do. We warmly invite the reviewer to respond and respectfully ask that they consider raising their score to recommend acceptance.
> > >
> > > _____
> > > [1] Frequency bands in RoPE, Oka et al., ICLR 2026
> > >
> > > [2] VideoRoPE, Zhang et al., ICML 2025
> > >
> > > [3] Circle-RoPE, Wang et al., NeurIPS 2025
> > >
> > > [4] Resonance RoPE, Zhu et al., ICML 2024
> > >
> > > [5] Group Representational Position Encoding, Zhang et al., ICLR 2026
> > >
> > > [6] Rethinking Attention with Performers, Choromanski et a;., ICLR 2021
> > >
> > > [7] Recipe for a General, Powerful, Scalable Graph Transformer, Rampášek et al., NeurIPS 2022 (please see esp. BigBird and Performer results)
> > >
> > > [8] Reconsidering Softmax and Linear Attention, Han et al., NeurIPS 2024
> > >
> > > [9] Simplex random features, Reid et al., ICML 2023

---

### Official Review · Reviewer_XxuS · 2026-03-17

**Soundness:** 3
**Presentation:** 3
**Significance:** 3
**Originality:** 3
**Overall Recommendation:** 5
**Confidence:** 4

**Summary:**

This paper presents provide WIRE, a positional encoding for graphs based on an extension of RoPEs to the graph domain. WIRE encodings avoid instantiating the full $N \times N$ attention matrix and are thus are compatible with linear attention. The authors first demonstrate how WIRE is obtained by applying Laplacian eigenvectors as RoPE inputs, show several desirable theoretical properties of the resulting encodings, and benchmark WIRE on a range of synthetic and real-world derived datasets to show tangible gains over the baseline linear-attention based Performer model.

**Compliance With Llm Reviewing Policy:**

Affirmed.

**Final Justification:**

The proposed method is simple and effective, and is motivated effectively both with empirical results and theoretical backing. The authors have also conducted a successful rebuttal and mainly addressed the remaining experimental concerns -- I particularly appreciate the late effort to compare to Reid et al. I am thus happy to increase my score and recommend acceptance.

**Key Questions For Authors:**

1. Per W1: Do the GNN benchmark experiments follow the synthetic tasks in Sec.. 4.1 that they have a similar injection of LapPE/RWSEs appended as node features (in addition to WIRE), in both WIRE/no WIRE models? The existence of “# Laplacian Eigenvectors” and “# RWSE Features” seems to indicate so, but this would benefit clarifying in the main text as well (Similar to in Sec. 4.1). In general, I ask the authors to clarify the ambiguities argued in W1.
2. Per W3: Can the authors provide any studies, or at the very least conjecture on how using (i) truncated eigenvectors and (ii) random sampling vs. learning frequencies affect the theoretical results on effective resistance and approximate gauge invariance properties?

**Limitations:**

The authors concede some limitations: Comparing with Graphormer is listed as future work; the assumptions made in theory and the gaps vs. practice are briefly mentioned.

Overall, this is a strong paper with a clear narrative and simple but effective proposed method. While I am leaning towards acceptance, some issues remain re: overall the experimental setup and important baselines not compared against. I deem W1 and W2 are more important to the core of the paper, while W3 is more of a welcome addition.

**Strengths And Weaknesses:**

**Strengths:**
1. WIRE is simple but effective — it is essentially a simple extension of RoPE by treating LapPEs as input coordinates, which is their primary use as PEs in the first place. The method has a small parameter cost, and has a clear, tangible efficiency benefit in compatibility with linear attention.
2. The simplicity of the method could have been interpreted as limiting the novelty of the work if presented as a standalone method contribution, but the authors provide a comprehensive theoretical analysis that demonstrates why & how RoPEs work well in the graph domain by (i) making clear connections to RoPE usage in sequential and grid-based domains, and (ii) demonstrating approximate convergence to effective resistance and approximate gauge invariance properties (under considerable assumptions, see W3).
3. The authors experiment on a wide array of settings: Synthetic and real-life derived graph benchmarks, point clouds, linear and softmax attention, as well as a variety of linear and $\mathcal{O}(N^2)$ transformers.


**Weaknesses:**
1. The experimental setup for the GNN benchmarks (Sec. 4.3) and its documentation, particularly regarding the use of PEs and identical settings in evaluation is murky in places and makes a fair and confident comparison difficult: In Appendix A.4.1, the following is mentioned: _“all the hyperparameter settings were identical to Rampasek et al. (2022), with optional 16 Laplacian positional encoding dimension for the WIRE Performer.”_ My understanding is that all GNN benchmark experiments compare a GraphGPS model with Performer attention & a GNN and hyperparameters as specified in Table 6, with 16 LapPEs are additionally fed to RoPE to produce WIRE, **but both the baseline and WIRE models are additionally provided PEs (LapPE and/or RWSE, as per Table 6) as node features, following GraphGPS.** Please clarify this (also see Q1). This ambiguity leads to the following (potential) issues:
   1. If the above interpretation that both baseline and WIRE models are provided PEs as node features, I assume this is in the spirit of benchmarking in a setting that is similar to how these models would be deployed in the real world — which is sensible. If my understanding above is wrong and the baseline does not have access to any PEs, that is problematic in that WIRE needs to be compared with other PEs in any case.
   2. The $\mathcal{O}(N^2)$ transformers are directly taken from Rampasek et al. (2022) which the authors do mention,  meaning a variety of different settings (which PEs are used and their dimensions) are used regarding the use of PEs. This doesn’t hurt the claims of this paper (if anything, this setup is advantageous for the $\mathcal{O}(N^2)$ transformer baseline), but makes it harder to make clear comparisons. I would also suggest the authors to differentiate between cases where WIRE matches/beats the $\mathcal{O}(N^2)$ transformer and doesn’t in Table 3 for better clarity.
   3. The PE usage should similarly be clarified for the SGFormer, BigBird and softmax transformer experiments in the appendix.
2. In any case, I think comparisons against a (linear) transformer that explicitly incorporates structural biases into the attention à la WIRE. Some potential avenues:
   1. Reid et al. (2024), which the authors cite, seem to provide a direct competitor to inject structural biases into linear transformers. They do not provide an implementation (which is problematic in itself), but the algorithm is pretty explicit and could be implemented. I do concede that this may result in substantial additional work.
   2. Assuming the authors do not go through with (1), I would like to see WIRE compared with other like attention-biasing methods that bias the attention mechanism (usually RPEs). I am aware that these forgo the linear attention, but also note that the authors themselves benchmark WIRE on $\mathcal{O}(N^2)$ transformers in a number of places (e.g. Table 8), so these benchmarks would be suitable to compare against other PEs on $\mathcal{O}(N^2)$ transformers.
      - The obvious choice would be Graphormer (Ying et al, 2021) or Exphormer here, Zhang et al. (2023) (which the authors also cite) also seems like a suitable candidate here.
      - An alternative could be adapting & naively injecting LapPE/RWSE-like encodings into the attention layer in similar fashion to WIRE.
3. As mentioned in S2, the authors demonstrate approximate convergence to effective resistance and approximate gauge invariance properties. While these theoretical discussions are useful to motivate the method and guide the narrative,how these approximations hold in practice is not addressed. Theorem 3 makes significant assumptions in using all $N - 1$ eigenvectors and randomly sampled frequencies. In practice, WIRE learns the frequencies and use the first $k$ eigenvectors. As a result, some sign ambiguity issues still remain, which are learned over training in GraphGPS-like fashion. Nevertheless, the practical size of this gap between the theory and practice is unclear, and additional studies on this would surely strengthen the theoretical motivation presented.

---

> ### Author Rebuttal · Authors · 2026-03-30
>
> We warmly thank the reviewer for their detailed reading of the text, and are pleased that they find the experiments diverse and the theoretical analysis comprehensive. We are happy to respond to all questions and concerns below.
>
> 1. **Confusing GNN benchmarks**. Thanks for the comments. We strongly agree that GNN benchmarking can sometimes be “murky”. In an attempt to make the cleanest comparison possible, we consider two settings — (1) adding WIRE to already strong GNN benchmarks *without any other modifications*, and (2) synthetic structural tasks tackled using simple, minimalist transformer architectures. As the reviewer says, setting (1) is intended to reflect realistic real-world applications and permits comparisons with numbers in other papers. Importantly, as the reviewer suggests, we do *not* remove existing PEs (aka LapPE/RWPE node features) in order to avoid artificially weakening the baselines. Our paper is interested in the gains provided by WIRE as an extra inductive bias of little cost — not so much as an alternative to other PEs. Conversely, setting (2) makes for a very straightforward ablation that can be replicated in a few lines of code. It is intended to *directly* demonstrate the structural inductive bias from WIRE on a task where structural information is key, without including all the other highly-optimised tricks one finds in eg GraphGPS (which we think may sometimes obfuscate findings). For SGFormer, BigBird and Softmax, we directly lift the most performant implementations from the respective papers and then simply add WIRE on top. We have now made sure all this is clearer in the paper. Thanks! (We have also made the cases where WIRE + Performer beats softmax visually clearer in the table).
> 2. **Other linear attention + WIRE baselines**. Thanks for the comment. We remark that there aren’t many linear attention graph PEs to compare against in the literature (hence this work), but we still strongly agree with the reviewer’s sentiment. In more detail:
> - *Reid et al.’s method*  [1]. Using GRFs does indeed achieve linear time complexity, by leveraging sparse linear algebra and a finite (probabilistic) receptive field. Whilst interesting, we respectfully suggest that this approach may not be directly comparable to WIRE — a simple, drop-in addition to existing *dense* models.
> - *RWSE encodings with WIRE*. We actually **already implement this idea in App A.5** -- see Tables 10 and 11. We also find gains, albeit in general smaller than those from WIRE with eigenvectors. We will be sure to flag this part of the paper more clearly because we agree that it may be helpful.
> - *Graphormer*. The reviewer correctly notes that bias-based RPEs like Graphormer rely on instantiation of attention, so they are not compatible with linear attention. Comparing *full rank* attention against WIRE + Performer may not make for a fair test. That said, we agree that including other benchmark results may still be helpful, so have added extra results in the style of Table 2 from 'Enhancing Graph Transformers' [2]. For fairness, we compare to *softmax* GPS + WIRE (Table 8). For instance, on CLUSTER, we have 78.02(6) with regular GPS, 78.19(2) with GPS + WIRE, 73.2(6) with GT, and 78.07(4) with Exphormer. The other results are omitted here due to the rebuttal character limit. **GPS is already a strong model compared to other dense baselines, and the boost from WIRE often makes it better still**. We think a full *theoretical* comparison with Graphormer -- which is expensive but exactly invariant wrt shortest path distance -- and WIRE -- which is cheaper and approximately invariant -- makes for interesting future work.
> 3. **Theorem 3 approximations**. As the reviewer suggests, Theorem 3 serves as an *existence proof* — it helps motivate why WIRE works well in practice, but it isn’t directly/exclusively responsible for the empirical gains we see in our experiments. Indeed, WIRE works well with truncated eigenvectors, learned frequencies, and even RWPE-based rotations, all of which lie beyond Thm 3’s assumptions. Naturally, learned weights, optimised for the particular task at hand, tend to work better for our tasks than random weights. We have made this important point clearer in the text; thanks. In applications where *exact* gauge invariance is essential, we recommend using RWPE coordinates in WIRE. (As an extra heuristic point: note that the spectral features in Thm 3 are normalised by the square root of the respective eigenvalues, so the truncated higher elements will tend to be smaller in norm and hence may contribute less to the effective resistance).
>
> We again warmly thank the reviewer, and invite them to respond. If satisfied with these clarifications and additions, we hope they will consider raising their score.
>
> [1] Linear Attention Topological Masking with Graph Random Features, Reid et al., ICLR 2025
>
> [2] Enhancing Graph Transformers with Hierarchical Distance Structural Encoding, Luo et al., NeurIPS 2024

---

> > ### Author Rebuttal · Reviewer_XxuS · 2026-04-04
> >
> > - **[W1]** Thanks for the clarifications. As mentioned, the current setup is sensible as long as it’s clarified in the paper accordingly.
> > - **[W3/Q3]** I appreciate the additional overview, and the explanation re: spectral truncation is sensible. I do not expect a major overhaul of the remaining “theory/practice gap” through the remaining rebuttal period and the provided studies are useful for the work, so I deem this point mostly resolved (even if partially).
> >
> > I thank the authors for their efforts in the rebuttal. The ambiguities re: W1 (which were crucial for a correct understanding of the evaluations) have been addressed sufficiently; they have also provided some additional support for W3/Q3. I do have some remaining concerns re: W2, though:
> > - I do not agree that Reid et al.’s method [1] is not directly comparable to WIRE. While their construction differs significantly, both are in essence $\mathcal{O}(N)$ structural attention methods for graphs, which is a key contribution of this work. The relative simplicity of WIRE may arguably be an advantage over Reid et al., but it does not absolve WIRE of a comparison with a directly competing, relevant and recent baseline.
> > - I welcome the additional results on GT and Exphormer, but will await for the omitted/further results by the end of the rebuttal deadline (incomplete/in progress results are fine given the timeline) before making my final decision.
> >
> > I thus maintain my score for the time being.

---

> > > ### Author Response · Authors · 2026-04-04
> > >
> > > Thanks for the acknowledgement, and for marking W1 and W3 resolved. Please find further clarifications re W2 below.
> > >
> > > *GT and Exphormer results.* With extra characters now available in this new response, the full GT and Exphormer results are as follows. (The GPS models here use softmax.)
> > >
> > > |  | GPS |  GPS + WIRE (ours)| GT | Exphormer |
> > > | -------- | ------- | ------- | ------- | ------- |
> > > | MNIST  | 98.05(4) | 98.46(3) | 90.8(1) | 98.55(4) |
> > > | CIFAR10 | 72.3(1) | 73.48(7) | 59.7(3) | 74.7(1) |
> > > | PATTERN | 86.69(2) | 86.75(2) | 84.8(1) | 86.74(2) |
> > > | CLUSTER | 78.02(6) | 78.19(2) | 73.2(6) | 78.07(4) |
> > > | ogbg-molhiv | 0.788(1) | 0.798(2) | - | - |
> > >
> > > GPS + WIRE consistently beats GT, and is on par with Exphormer (both perform best for two datasets). However, we caution the following when interpreting the table: **WIRE is an *extra* structural inductive bias for attention layers, that could equally be incorporated into GT or Exphormer**. For instance, it is very plausible that Exphormer + WIRE would also be better than Exphormer (but beyond the scope of this rebuttal period). This may be a better comparison than GPS + WIRE vs Exphormer. Nonetheless, we agree the extra baselines may be interesting so have added them to Table 8.
> > >
> > > *RWPE WIRE*. Just checking you've seen that we **already include the RWPE WIRE experiments which you requested in your earlier review**, in Tables 10 and 11 of our paper. We hope these results are helpful; please let us know if you have any questions.
> > >
> > > *Graph random features (GRFs)* [1]. We remain sceptical that GRFs are a suitable baseline for this setting; they do not involve global attention, which is well known to be important for these GNN tasks [2], and the authors prefer to apply GRFs to vision/robotics datasets which are quite different to the setting in our paper. **We also again stress that GRFs could straightforwardly be used *with* WIRE because WIRE is an extra inductive bias for attention**, so in practice one should really compare GRFs + WIRE vs pure GRFs. As such, respectfully, we do not consider GRFs to be a 'directly competing, relevant and recent baseline' for this paper.
> > >
> > > Notwithstanding the above, we will try to run some GRFs for GNN benchmarks during the next few days. The setup is nontrivial because GRFs have not previously been used for these tasks and of course time is now very limited. We will update the reviewer as we can.
> > >
> > > Thanks once more for the thorough review and engaging discussion. We look forward to hearing your thoughts on the above!
> > >
> > > ___
> > >
> > > **EDIT 04/04/26**: Following some extra last-minute experiments by one of the authors, we are happy to include the following **additional results for GNN benchmarks with GRFs**. Here, the GPS models use Performer attention. We concatenate queries and keys with Reid et al's graph random features for an extra structural inductive bias.
> > >
> > > |  | GPS |  GPS + WIRE (ours) | GPS + GRFs [1] |
> > > | -------- | ------- | ------- | ------- |
> > > | MNIST  | 97.56 | 98.10 | 97.85 |
> > > | CIFAR10 | 70.61| 71.15 | 70.65 |
> > > | PATTERN | 85.71| 86.63 | 85.86 |
> > > | CLUSTER | 76.90| 77.53 | 76.87 |
> > >
> > > As anticipated, in this setting the gains provided by augmenting attention with GRFs are smaller than the gains provided by WIRE. We have added this to the manuscript; thanks.
> > >
> > > We look forward to hearing your thoughts on the above. Having incorporated this final suggestion, very much hope that you will consider raising your score.
> > > ___
> > >
> > > [1] Linear Attention Topological Masking with Graph Random Features, Reid et al., ICLR 2025
> > >
> > > [2] Exphormer: Sparse transformers for graphs, Shirzad et al., ICML 2023

---

### Official Review · Reviewer_k9zE · 2026-03-18

**Soundness:** 4
**Presentation:** 4
**Significance:** 3
**Originality:** 3
**Overall Recommendation:** 5
**Confidence:** 4

**Summary:**

RoPE is a positional encoding used in LLMs and ViTs. Its performance is tested with graph structured data. they introduce wire a rope-style encoding, show it improves transformer performance on many benchmark tasks. The wire PE is added to a transformer with linear attention to allow to hold full adjacency information in memory and save memory resources. The benchmark models are transformers with linear attention and a transformer with exponential attention. the transformer with wire outperforms both on all tasks.

**Compliance With Llm Reviewing Policy:**

Affirmed.

**Key Questions For Authors:**

Whenever I see performance comparison, on data I am not familiar with, I struggle to interpret the significance of the scale. It would be useful to have a history of the recent improvements for popular benchmarks in order to judge where we stand on the "meta learning curve". I do not expect the authors to make this addition, rather I want to share a thought I have upon gazing at strange numbers.

Please explain what were the challenges in adapting RoPE for I am not familiar with RoPE, do you have an idea why it is successful?
proof of theorem 1, in "... the Laplacian is derived directly from the adjacency matrix so it is permutation equivariant" "derived" is vague.

minor: I assume a "performer" is a transformer with linear attention, this should be explicitly stated around section 4.3.

**Limitations:**

Why was WIRE not used on exponential attention transformer?

**Strengths And Weaknesses:**

The paper does useful work of transposing the RoPE positional encoding to a different setting. The presentation is clear and the work is generally satisfactory.

---

> ### Author Rebuttal · Authors · 2026-03-30
>
> We warmly thank the reviewer for their positive comments on the text. We are pleased that they find the presentation clear. All questions and concerns are addressed in detail below.
>
> 1. **“Where are we on the meta learning curve?”** Thanks for the question. We agree that graph benchmarks can be difficult to interpret, and hence tend to prefer our structural synthetic tasks where the benefits of WIRE are obvious (Sec 4.1). For the reviewer’s reference, in the context of the already heavily optimised benchmarks in Sec 4.3, our observed gains of several percentage points are generally considered fair/strong — especially for such a simple, cheap drop in algorithmic addition. See e.g. Table 3 of GraphGPS [1], where the authors’ model often beats baselines by very narrow margins (and is actually not consistently best), or Table 2 of SGFormer [2]. With this in mind, we have now also added extra historical results to the appendix to make the 'meta learning curve' clearer. Thanks.
> 2. **“What are the challenges with applying RoPE to graphs? Why is it successful?”** The key challenge we encounter is that, unlike text,  graphs lack a canonical coordinate system to plug into RoPE. We show that one particular choice of node “coordinates” — the Laplacian eigenvectors — gives (asymptotic) dependence on effective resistance (Thm 3). We find that this provides a strong inductive bias that helps boost performance on structural tasks. We think this helps explain why our algorithm is successful, and agree that further investigation is an important direction for future work.
> 3. **“The Laplacian is derived from the adjacency matrix so it is permutation equivariant”**. Thanks for flagging this confusion; please allow us to make this statement more precise. Let $\mathbf{L} = \mathbf{D} - \mathbf{A}$ be the graph Laplacian, where $\mathbf{D}$ is the degree matrix and $\mathbf{A}$ is the adjacency matrix. A permutation $\mathbf{P}$ of the node indices gives a permuted Laplacian $\mathbf{PLP}^{\top}$, since the rows and columns of the matrix are both permuted. The eigenvectors of this permuted Laplacian are trivially $\mathbf{Pu}$, where $\mathbf{u}$ is an eigenvector of $\mathbf{L}$, with the eigenvalues unchanged. Hence, a PE using these eigenvectors as features must be equivariant under node order permutation.
> 4. **Performer vs linear attention**. We use “Performer” and “linear attention” interchangeably in this paper. We have updated the text to make this clearer. Thanks.
> 5. **“Why was WIRE not used on exponential attention transforme?”** In fact, we **do already apply WIRE with softmax in several places in our paper**, in many cases observing strong gains — see e.g. Tables 1, 2, 4 and 8. Note that we omit some bigger datasets from Table 8 compared to the linear attention experiments to save compute. Having already trained >200 models in total, we respectfully suggest that the trend of interest -- namely, WIRE also improving transformers using softmax -- may already be clear.
>
> We again sincerely thank them to reviewer for reading the text, and invite them to respond with any further questions.
>
> [1] Recipe for a General, Powerful, Scalable Graph Transformer, Rampášek et al., NeurIPS 2022
>
> [2] SGFormer: Simplifying and Empowering Transformers for Large-Graph Representations, Wu et al., NeurIPS 2023

---

### Decision · Program_Chairs · 2026-04-30

**Decision:**

Accept (spotlight)

**Comment:**

In this paper, the authors introduce and analyse a new Positional Encoding (PE) for graphs, inspired by rotary PE for LLMs. The reviewers were globally positive or very positive, with many questions and suggestions that were answered by the authors to satisfaction. One small concern was raised to improve the motivation of the paper, but the consensus view of the paper remains positive.